# Preparation of Allyl Alcohol Oligomers Using Dipicolinate Oxovanadium(IV) Coordination Compound

**DOI:** 10.3390/ma15030695

**Published:** 2022-01-18

**Authors:** Kacper Pobłocki, Dagmara Jacewicz, Juliusz Walczak, Barbara Gawdzik, Karol Kramkowski, Joanna Drzeżdżon, Paweł Kowalczyk

**Affiliations:** 1Department of Environmental Technology, Faculty of Chemistry, University of Gdansk, Wita Stwosza 63, 80-308 Gdansk, Poland; kacperpoblocki@wp.pl (K.P.); dagmara.jacewicz@ug.edu.pl (D.J.); 2Department of Organic Chemistry, Faculty of Chemistry, Gdansk University of Technology, Narutowicza 11/12, 80-233 Gdansk, Poland; juliusz.walczak@pg.edu.pl; 3Institute of Chemistry, Jan Kochanowski University, Uniwersytecka 7, 25-406 Kielce, Poland; b.gawdzik@ujk.edu.pl; 4Department of Physical Chemistry, Medical University of Bialystok, 15-089 Bialystok, Poland; kkramk@wp.pl; 5Department of Animal Nutrition, The Kielanowski Institute of Animal Physiology and Nutrition, Polish Academy of Sciences, Instytucka 3, 05-110 Jabłonna, Poland

**Keywords:** oligomerization, oxovanadium(IV) complexes, 2-propen-1-ol, catalysis, mechanism of oligomerization, dipicolinate anion

## Abstract

Currently, new precatalysts for olefin oligomerization are being sought in the group of vanadium(IV) complexes. Thus, the aim of our research was to examine the catalytic activity of the oxovanadium(IV) dipicolinate complex [VO(dipic)(H_2_O)_2_] 2 H_2_O (dipic = pyridine-2,6-dicarboxylate anion) in 2-propen-1-ol oligomerization as well as to characterize oligomerization products using matrix-assisted laser desorption/ionization–time-of-flight mass spectrometry (MALDI-TOF-MS), infrared spectroscopy (IR) and nuclear magnetic resonance (NMR). The oligomerization process took place at room temperature, under atmospheric pressure and under nitrogen atmosphere to prevent oxidation of the activator MMAO-12—the modified methylaluminoxane (7 wt.%) aluminum in toluene. The last point was to determine the catalytic activity of the complex in the oligomerization reaction of 2-propen-1-ol. The aspect that enriches this work is the proposed mechanism of oligomerization of allyl alcohol based on the literature.

## 1. Introduction

When you hear “polymer”, the first thing that comes to mind is plastics. However, increasingly in publications, authors write about polymers or oligomers not only meaning their use in the production of car tires [1,2], packaging [3,4], foil [5,6] or oil [7,8], but also polyolefins used in the production of medical implants [9,10], anti-HIV (human immunodeficiency virus) therapy [11,12], green chemistry [12,13,14] and Alzheimer’s treatment [15,16]. The synthesis of polymers requires special conditions, therefore, catalysts are used which lower the activation energy and speed up the process [17,18,19,20,21,22,23]. It has become popular to use metallocenes, e.g., complex compounds containing d-block metals and organic ligands (precatalyst) [20,21]. The combination of a precatalyst with an activator, i.e., an organoaluminum compound, e.g., methylaluminoxane (MAO) or a modified methylaluminoxane (MMAO-12, 7% aluminum in toluene) creates a Ziegler–Natta catalyst [24,25,26,27]. The sixth generation of Ziegler–Natta catalysts is the most widespread due to high catalytic activity and attempts to replace MMAO with another activator. The reason is that MMAO changes its structure and composition during storage. Modified methyl aluminoxane, as an activator, oxidizes very quickly when there is oxygen in the reaction system and, therefore, nitrogen is introduced to prevent this.

The subject of interest of scientists at the beginning of the 20th century was the comparison of the *trans* spatial structure [VO(dipic)(H_2_O)_2_] 2 H_2_O (1) to the known compound [VO(dipic)(o-phen)] 3 H_2_O (2) by X-ray crystallography and to obtain (2) from (1) by substituting two water molecules with 1,10-phenanthroline. It turned out that the coordination sphere around the oxovanadium(IV) ion was completely transformed during the reaction, which influenced the kinetic aspect [28]. The oxovanadium(IV) dipicolinate complex compound, in its structure, contains dipic (dipicolinate anion), which acts as a tridentate ligand. Thanks to the free electron pair on the nitrogen atom, it can form stable chelates with cations of oxometals from block-d, while showing very different coordination properties. It is used to remove corrosion, decontaminates nuclear reactors, and takes part in biological processes as a carrier of electrons and medical bioimaging [29,30].

Oxovanadium(IV) compounds are used as precatalysts in the polymerization of olefins due to their high catalytic activity and the quality of the products obtained. Vanadium complex compounds are used as catalysts in industrial production of synthetic rubbers, elastomers and polyethylene [31]. However, in our case, special attention was given to the dipicolinate complex of oxovanadium(IV), due to its widely described physicochemical and biological properties such as combating diabetes type I and II [32], cell metabolism [33], antioxidant properties [34], plasmid DNA cleavage, chromosomal aberrations and use in anticancer therapy [35,36]. Thorn et al. have reported the application of V-dipic complexes as: [VO(dipic)(i-PrO)], [VO(dipic)(pinme)], [VO(dipic)(dpheol)] and analogs in stoichiometric aerobic oxidation of isopropanol and other alcohols as lignin models [37]. Another example is Gawdzik et al. who reported new oxovanadium(IV) microclusters with 2-phenylpyridine which showed highly activity for the 3-buten-1-ol, 2-chloro-2-propen-1-ol, allyl alcohol, and 2,3-dibromo-2-propen-1-ol oligomerizations [38].

In this publication, for the first time the dipicolinate complex of oxovanadium(IV) is presented as a new precatalyst for an olefin oligomerization. We examined its catalytic properties in the oligomerization of allyl alcohol. The oligomerization reaction products were also analyzed using mass spectrometry techniques such as matrix-assisted laser desorption/ionization time of flight mass spectrometry (MALDI-TOF-MS), infrared spectroscopy (IR) and nuclear magnetic resonance (NMR). The oligomerization process took place at room temperature, under atmospheric pressure and under nitrogen atmosphere, so that the activator, which was MMAO-12, would not oxidize [39]. The aspect that enriches this work is the proposed mechanism of oligomerization of allyl alcohol based on the literature.

## 2. Materials and Methods

### 2.1. Materials

All chemical compounds (vanadyl acetylacetonate, dipicolinic acid, modified methylaluminoxane (7% aluminum in toluene), 2-propen-1-ol) used in this work were purchased from Sigma-Aldrich (Darmstad, Germany). Their purity was between 98% and 100%.

### 2.2. Dipicolinate Oxovanadium(IV) Complex Synthesis

Aqueous vanadyl acetylacetonate (VO(acac)_2_) (2.13 mmol, 0.57 g) was added to dipicolinic acid (H_2_dipic) (2.15 mmol, 0.36 g). Then 50 cm^3^ of water was added to the mixture. The entire solution was heated at 100 °C to reflux for 90 min until the mixture changed color. It took 30 min to cool down. One month later, the dipicolinate oxovanadium(IV) complex crystallized in the solution in the form of blue crystals. Crystallization was carried out at room temperature. Crystallization lasted so long that it prevented the introduction of impurities that could appear if the process was carried out under conditions of reduced temperature.

### 2.3. Elemental Analysis of the Oxovanadium(IV) Complex Compound with Pyridine-2,6-Dicarboxylate Anion

Elemental analysis of the complex was performed with the Vario El Cube apparatus. The samples tested by means of elemental analysis were dry and homogeneous with a mass of 2 mg.

### 2.4. Infrared (IR) Spectra

The examination of the the oxovanadium(IV) complex compound with pyridine-2,6-dicarboxylate anion and the oligomerization product by infrared spectroscopy (IR) was performed in the range from 4000 cm^−1^ to 600 cm^−1^ on a KBr pastil. The measurement was carried out on a Bruker IFS 66 spectrometer (Evisa, Tucson, AZ, USA). The IFS 66 apparatus performed infrared spectra in the Fourier transform with a resolution of 0.12 cm^−1^. DLATGS was a detector in IR measurements.

### 2.5. Matrix-Assisted Laser Desorption/Ionization Time of Flight Mass Spectrometry (MALDI-TOF-MS) Spectra

Molecular weights of the 2 propen-1-ol oligomer chains were determined using MALDI-TOF-MS spectrometer from the Bruker Biflex III company (Billerica, MA, USA). 2,5-Dihydroxybenzoic acid (DHB) was used as a matrix.

### 2.6. Nuclear Magnetic Resonance (NMR) Spectra

Nuclear magnetic resonance spectra of oligomerization products were recorded with a Bruker Avance III 500 spectrometer (Billerica, MA, USA). The measurement was carried out at 25 °C. The measured frequency was 126 MHz for ^13^C NMR and 500 MHz for ^1^H NMR. The solvent that was used was deuterated 1,1,2,2-tetrachloroethane.

### 2.7. The Oligomerization Process

The oligomerization process was carried out in a glass flask closed with a stopper (Figure 1). First, the precatalyst which was [VO(dipic)(H_2_O)_2_] 2 H_2_O (Figure 2) (3 μmol, 0.912 mg) was dissolved in 1 mL of toluene and 1 mL of anhydrous DMSO (anhydrous dimethyl sulfoxide). The solution was then mixed with a magnetic stirrer. In the next step, the following reagents were added: 3 mL of MMAO-12 (modified methylaluminoxane, 7% aluminium in toluene) and 3 mL of 2-propen-1-ol. The whole oligomerization process was undertaken at ambient pressure (1013 hPa), at room temperature and in nitrogen air. After 90 min of oligomerization, a white gel was obtained and then it was washed with a mixture of 1 M hydrochloric acid and 1 M methanol in a 1:1 molar ratio.

## 3. Results and Discussion

The structure of the complex is well known and described in the literature [40]. Table 1 shows the percentages of elements obtained by exploiting the elemental analysis (AE) technique and theoretical calculations.

Theoretical data have been calculated according to the following procedure:

Mass of [VO(dipic)(H_2_O)_2_] 2 H_2_O = 305.9 g/mol; %C = 84/305.9 × 100% = 27.46%; %H = 13/305.9 × 100% = 4.25%; %N = 14/305.9 × 100% = 4.58%.

In order to confirm the structure of the synthesized crystal of complex compound, we described the IR spectrum of the dipicolinate oxovanadium(IV) complex (Figure 3) [41].

The characterization results presented in Table 1 and Table 2 and their validation with theoretical calculations and literature data indicate that the vanadium complex synthesis was correct [41,42]. The synthesized complex was used for 2-propen-1-ol oligomerization. Analysis of the IR spectrum (Figure 4) of the product of 2-propen-1-ol oligomerization is presented in Table 3. The absorption of infrared radiation is accompanied by changes in the vibrational energy of the molecules. Since this energy is quantified, only radiation with certain energies, specific to the functional groups performing the vibrations, is absorbed. This makes it possible to determine which functional groups are present in the analyzed sample. The condition for absorption of radiation is the change in the dipole moment of the molecule during the process. The results of the IR studies showed that the end product of the oligomerization contained a double bond and a hydroxyl group. The IR studies confirmed the structure of the oligomerization products [43,44].

Using the MALDI-TOF-MS method, we characterized certain peaks, thus allowing the identification of the number of units present in the oligomer chains using [VO(dipic)(H_2_O)_2_] 2 H_2_O as a precatalyst (Figure 5).

The presence of oligomer chains of a specific length was confirmed by using mass spectrometry. The appropriate units were assigned to the peaks in the spectra formed in the process of 2-propen-1-ol oligomerization catalyzed by [VO(dipic)(H_2_O)_2_] 2 H_2_O. The 649.9 *m*/*z* peak was derived from 2,5-dihydroxybenzoic acid—the matrix and molecular peak were identified with a mass/charge ratio of 703.9 *m*/*z* that contained 2-propen-1-ol 12 units. The next peaks at *m*/*z* = 876.9 (15 units), *m*/*z* = 1066 (18 units) were observed in the attached mass spectrum. This was a confirmation that in the obtained 2-propen-1-ol mixture, the oligomers contained chains consisting of 12, 15 and 18 allyl alcohol units.

Analysis of 2-propen-1-ol oligomerization products was conducted using nuclear magnetic resonance spectroscopic techniques. The ^1^H NMR spectrum is shown in Figure 6 and the ^13^C NMR spectrum is shown in Figure 7.

In order to illustrate the structure of 2-propen-1-ol oligomers more precisely, the Table 4 and Table 5 based on the ^1^H NMR and ^13^C NMR spectra were prepared. The peak values corresponding to specific carbon and hydrogen atoms depending on the type of spectrum have been highlighted. NMR spectroscopy is based on the observation of transitions between magnetic energy levels of the ^1^H hydrogen isotope in the case of ^1^H NMR. A lot of information about the structure of the molecule under study can be obtained from the NMR spectra. The number of signals provides information about the number of protons lying in the same environment. The intensity of the signals is proportional to the number of protons associated with this signal. On the other hand, the values of chemical shifts of signals in the spectrum depend on the environment in which the protons are located. The larger the peak, the stronger the coupling of the interaction between adjacent electron nuclei, the so-called spin-spin couplings. NMR and IR test results confirm the structure of the oligomers obtained consisting of linked units of allyl alcohol.

The catalytic activity (Ca) for the [VO(dipic)(H_2_O)_2_] • 2 H_2_O complex compound can be calculated from the formula:(1)$$C_{a}=\frac{m}{n\cdot p\cdot t}=191.53\;\frac{g}{\mathrm{mmol}\cdot \mathrm{bar}\cdot h}$$
where: m—mass of obtained oligomer [g]; *n*—number of mmoles of V^4+^ [mmol]; *p*—pressure [bar]; *t*—oligomerization time [h].

The literature values were compared in Table 6 with the result of the calculations to find out how effective the precatalyst was. The number of mmole of V^4+^ used to calculate the catalytic activity (Ca) was calculated theoretically. In these calculations there is the lack of mass balance calculation, thus the calculation has uncertainties that can lead to misleading comparisons with the data presented in Table 6.

Comparing the calculated results with the literature values, we concluded that the precatalyst [VO(dipic)(H_2_O)_2_] 2 H_2_O belonged to the group of catalysts with high catalytic activity. The highest catalytic activity in the research to date had been noticed with olefins containing chlorine in their structure, for example 2-chloro-2-propen-1-ol. Thus, by achieving high purity, process efficiency could be increased. The use of ligands also played an important role. Too extensive chains of organic clusters caused steric barrier and thus low selectivity.

## 4. The Proposed Mechanism of the Oligomerization Reaction of 2-Propen-1-ol Catalyzed by [VO(dipic)(H_2_O)_2_] 2 H_2_O + MMAO-12

The mechanism of the oligomerization reaction of 2-propen-1-ol catalyzed by [VO(dipic)(H_2_O)_2_] 2 H_2_O, with the participation of an activator (MMAO-12), followed the mechanism of coordination polymerization based on the literature [46,47,48,49] (Figure 8). The approach of allyl alcohol to the center of oxovanadium(IV) with the participation of MMAO-12 caused the formation of the π complex between the alcohol’s double bond and the active center (Step 1). In the next step, coupling took place between the terminal carbon atom and the active center of oxovanadium(IV), at the expense of the water molecule from the precatalyst, which migrated to the activator. The resulting electrophilic center—carbocation—then underwent a nucleophilic attack by the π bond of allylic alcohol and thus propagated the alkyl chain (Step 2). In the elimination stage, the obtained oligomer separated from the dipicolinate complex of oxovanadium(IV) and modified methylaluminoxane, through the migration of a water molecule from MMAO-12 to the active center (Step 3). Washing the oligomer with a mixture of diluted hydrochloric acid and methanol removed residual catalyst and activator. The washing step was important because the activator also reverted to its native form and the cleaved oligomer, due to the momentary and opposite polarity on the carbon atoms, could produce a polymeric structure written according to the standard that could restore the stable structure of the catalyst under hydrolytic conditions. Another fact confirmed the correctness of the proposed termination step, involving the experiment, in which we added water to the reaction mixture after the oligomerization process. As a result, we observed a free activator molecule precipitating out from a solution, suggesting that it was not bound to the catalyst any more. It is worth emphasizing that this mechanism was also based on the fact that the greater the weight of a polymer, the better the properties as a leaving group it constituted, thus making this reaction self-limiting [46,47,48,49].

## 5. Conclusions

The oxovanadium(IV) dipicolinate complex compound is an effective precatalyst for the oligomerization process of 2-propen-1-ol carried out at room temperature, under nitrogen atmosphere and at atmospheric pressure. Several methods—IR, MALDI-TOF-MS, ^1^H and ^13^C NMR—confirmed that the obtained mixture of 2-propen-1-ol oligomers contained 12, 15 and 18 units of allyl alcohol. It can be stated that the dipicolinate complex of oxovanadium(IV) was a highly active precatalyst with 191.53 g mmol^−1^ bar^−1^ h^−1^ catalytic activity value. Our results showed that further examinations towards the potential olefin oligomerization precatalysts need to be undertaken, especially those derived from the dipocolinate oxovanadium(IV) complex.

## Figures and Tables

**Figure 1 materials-15-00695-f001:**
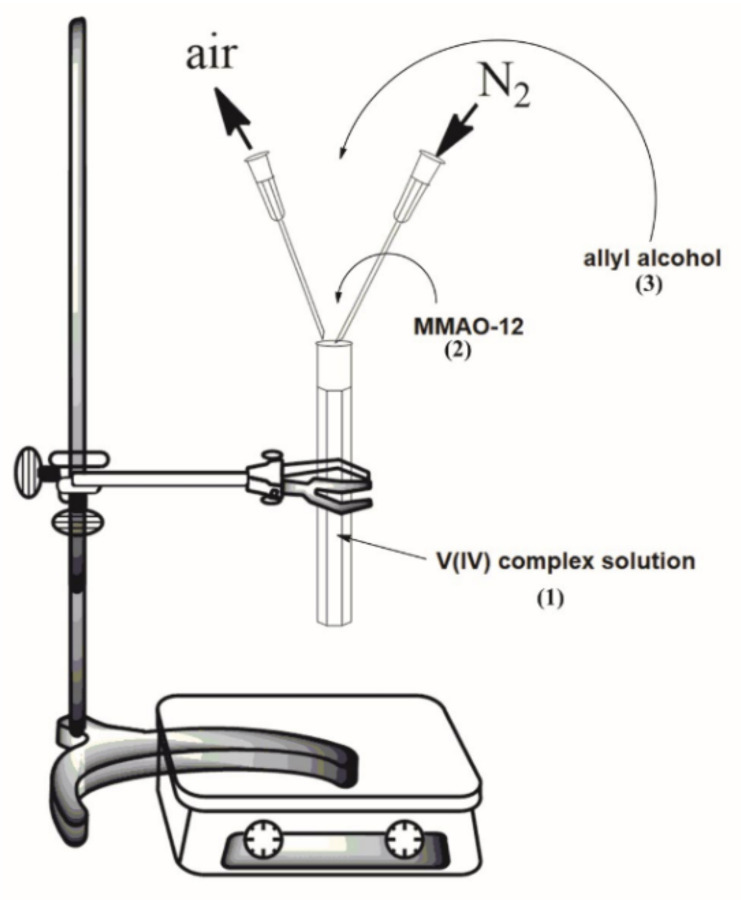
A process flow diagram for the oligomerization system.

**Figure 2 materials-15-00695-f002:**
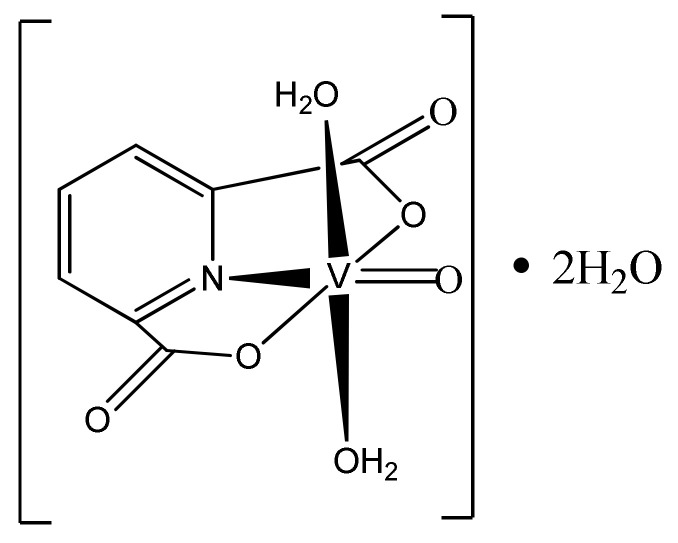
Chemical structure of [VO(dipic)(H_2_O)_2_] 2 H_2_O.

**Figure 3 materials-15-00695-f003:**
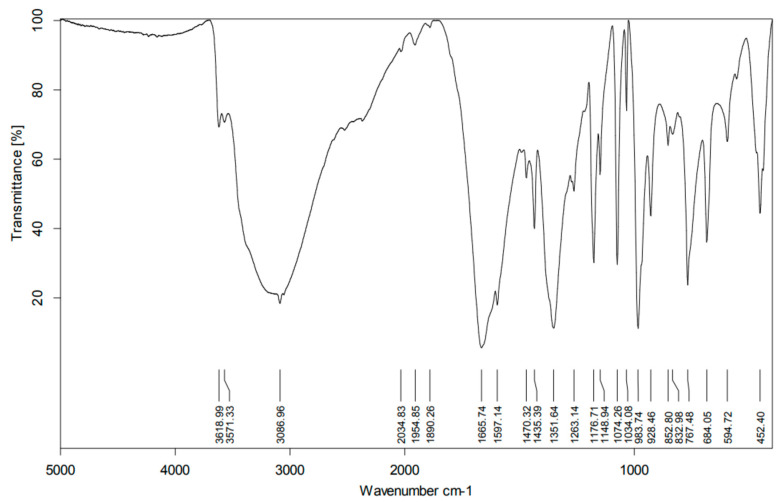
Infrared (IR) spectrum of [VO(dipic)(H_2_O)_2_] 2 H_2_O [41].

**Figure 4 materials-15-00695-f004:**
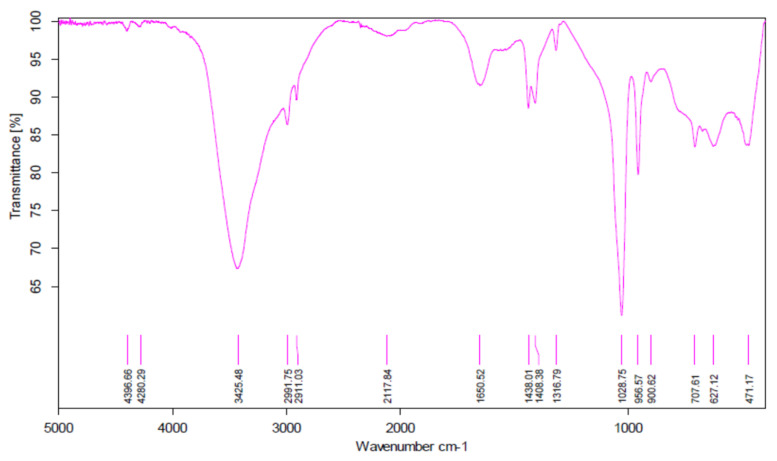
IR spectrum of the 2-propen-1-ol oligomerization product.

**Figure 5 materials-15-00695-f005:**
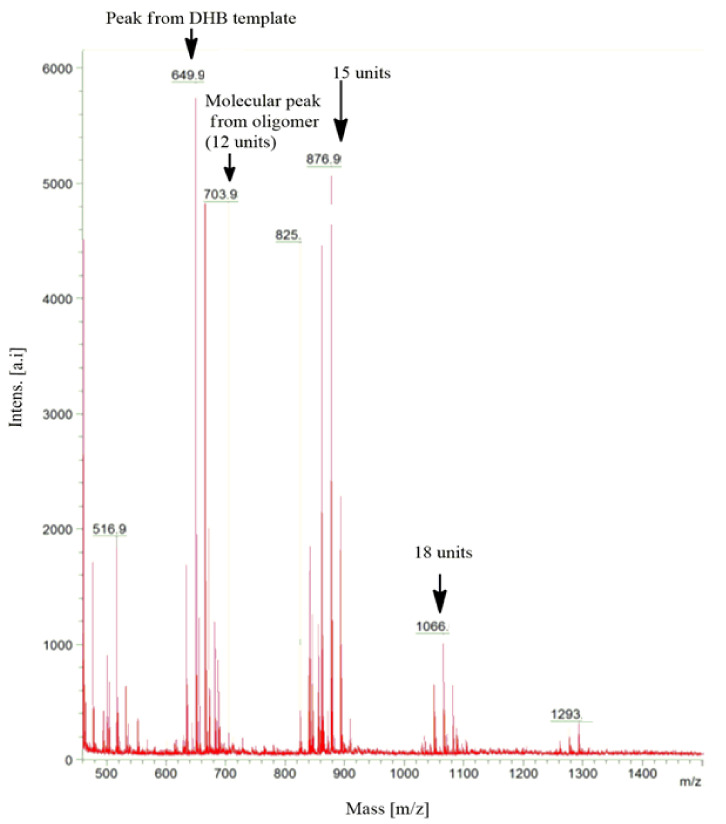
The matrix-assisted laser desorption/ionization time of flight mass spectrometry (MALDI-TOF-MS) spectrum of the products of the oligomerization process.

**Figure 6 materials-15-00695-f006:**
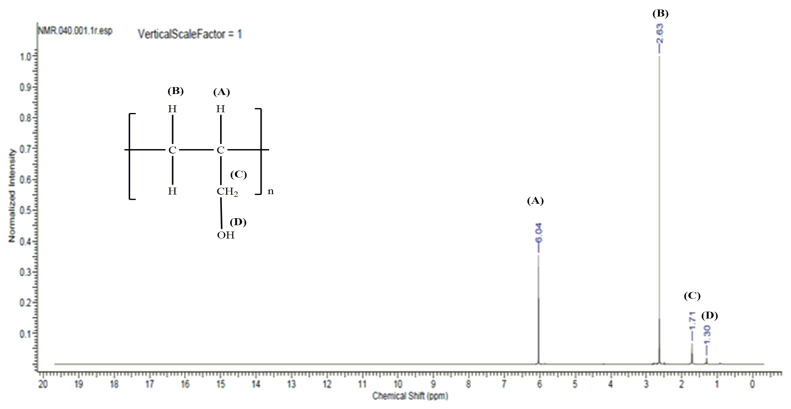
^1^H nuclear magnetic resonance (NMR) spectrum for 2-propen-1-ol oligomerization products obtained with the application of [VO(dipic)(H_2_O)_2_] • 2 H_2_O + MMAO-12.

**Figure 7 materials-15-00695-f007:**
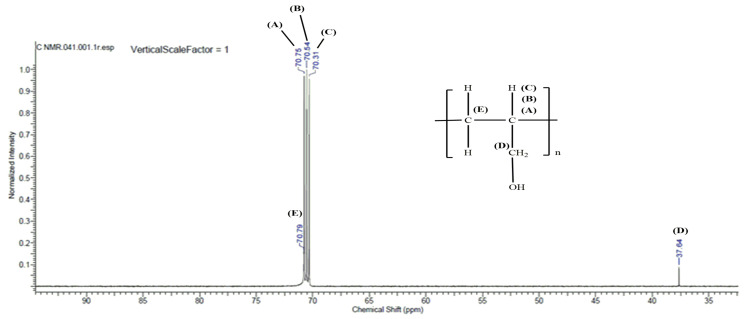
^13^C NMR spectrum for 2-propen-1-ol oligomerization products obtained with the application of [VO(dipic)(H_2_O)_2_] 2 H_2_O + MMAO-12.

**Figure 8 materials-15-00695-f008:**
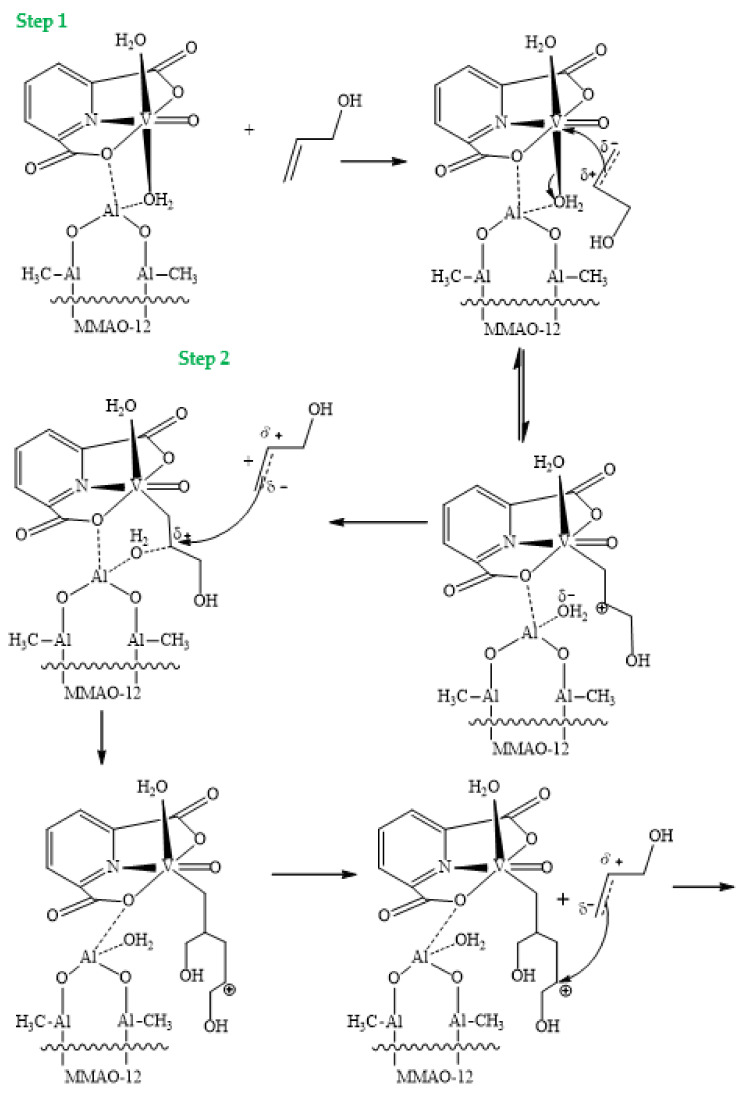
The proposed mechanism of 2-propen-1-ol oligomerization with the application of [VO(dipic)(H_2_O)_2_] 2 H_2_O + MMAO-12. Step 1 (Initiation), Step 2 (Propagation), Step 3 (Termination) [46,47,48,49].

**Table 1 materials-15-00695-t001:** Results of elemental analysis of the synthesized complex [VO(dipic)(H_2_O)_2_] • 2H_2_O (AE means experimental data, T denotes theoretical data).

Complex Compound	Percentage [%]					
	%C		%H		%N	
	AE	T	AE	T	AE	T
[VO(dipic)(H_2_O)_2_] 2 H_2_O	27.64	27.46	3.60	4.25	4.70	4.58

**Table 2 materials-15-00695-t002:** Characteristic IR spectrum absorption bands of [VO(dipic)(H_2_O)_2_] 2 H_2_O [41,42].

Wavenumber [cm^−1^]	Type of Vibration with Function Group
3571	*v*(OH)
1665	*v*(COO) of dipic
1352	*v*(COO) of dipic
983	V=O stretching frequency
452	stretching vibration of the V-N

**Table 3 materials-15-00695-t003:** Characteristic IR spectrum absorption bands for the 2-propen-1-ol oligomerization product [43,44].

Wavenumber [cm^−1^]	Type of Vibration	Function Group
3425	stretching vibrations	−OH
2992	stretching vibrations	−CH
1651	stretching vibrations	C=C
1438	bending vibrations	−CH_2_

**Table 4 materials-15-00695-t004:** Peak values and the corresponding hydrogen atoms derived from the 1H NMR spectrum for the products of 2-propen-1-ol oligomerization catalyzed by [VO(dipic)(H2O)2] • 2 H2O + MMAO-12.

Peak Value	Assigned Hydrogen Atoms
6.04	CH_2_=CH- (oligomer)
2.63	CH_2_=CH- (monomer)
1.71	HO-CH_2_- (oligomer)
1.30	-OH (oligomer)

**Table 5 materials-15-00695-t005:** Peak values and the corresponding hydrogen atoms derived from the ^13^C NMR spectrum for the products of 2-propen-1-ol oligomerization catalyzed by [VO(dipic)(H_2_O)_2_] 2H_2_O + MMAO-12.

Peak Value	Assigned Carbons Atoms
70.79	HO-CH_2-_CH-CH_2_- (oligomer)
70.76–70.31	HO-CH_2-_CH-CH_2_- (oligomer)
37.64	-CH_2_-OH (oligomer)

**Table 6 materials-15-00695-t006:** Catalyst efficiency classification based on their catalytic activity [45].

Catalyst Efficiency	Catalytic Activity [g∙mmol^−1^∙bar^−1^∙h^−1^]
Very low	<1
Low	1–10
Moderate	10–100
High	100–1000
Very high	>1000

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
