# Peer review of "Preparation of Allyl Alcohol Oligomers Using Dipicolinate Oxovanadium(IV) Coordination Compound"

_materials, 2022, doi:10.3390/ma15030695_

Round 1
Reviewer 1 Report
- Page 1 Lines no 7-19 have unequal tab spacing with margin.
Abstract
- Page 1 Line no 27, “Thus, the aim of our research was the examination of catalytic activity the oxovanadium(IV) dipicolinate complex [VO(dipic)(H2O)2].” may be written as “Thus, the aim of our research was the examination of catalytic activity of the oxovanadium(IV) dipicolinate complex [VO(dipic)(H2O)2].”
- Page 1 Line no 32, “The oligomerization process took place at room temperature, under atmospheric pressure and under nitrogen atmosphere to prevent oxidation of the activator MMAO‑12.” The full form of MMAO-12 should be mentioned in the abstract.
Introduction
- Page 2 Line no 42, “When you hear "polymer", the first thing comes to your mind are plastics?” may be written as” When you hear "polymer", the first thing comes to your mind is plastics?”
- Page 2 Line no 49, “It has become popular to use metallocenes, i.e complex compounds containing d‑block metals and organic ligands (precatalyst) [20, 21].” may be written as “It has become popular to use metallocenes, e. complex compounds containing d‑block metals and organic ligands (precatalyst) [20, 21]."
- Page 2 Line no 56, “Modified methyl aluminoxane, as an activatoroxidizes very quickly when there is oxygen in the reaction system, therefore nitrogen is introduced to prevent this.” may be written as “Modified methyl aluminoxane, as an activator, oxidizes very quickly when there is oxygen in the reaction system therefore, nitrogen is introduced to prevent this. “
- Page 2 Line no 59, “The subject of interest of scientists at the beginning of the 20th century was the comparison of the trans spatial structure [VO(dipic)(H2O)2] • 2 H2O (1) to the known compound [VO(dipic)(o‑phen)] • 3 H2O (2) by X‑ray crystallography and obtaining (2) from (1) by substituting two water molecules with 1,10‑phenanthroline.” may be written as “The subject of interest of scientists at the beginning of the 20th century was the comparison of the trans spatial structure [VO(dipic)(H2O)2] • 2 H2O (1) to the known compound [VO(dipic)(o‑phen)] • 3 H2O (2) by X‑ray crystallography and to obtain/obtained (2) from (1) by substituting two water molecules with 1,10‑phenanthroline.”
- Page 2 Line no 62, “It turned out that the coordination sphere around the oxovanadium(IV) ion was completely transformed during the reaction, which influences the kinetic aspect [28].” may be written as “It turned out that the coordination sphere around the oxovanadium(IV) ion was completely transformed during the reaction, which influenced the kinetic aspect [28].”
- Page 2 Line no 68, “It is used to remove corrosion, decontaminates nuclear reactors, and takes part in biological processes as a carrier of electrons and medical bioimaging [29, 30].” may be written as “It is used to remove corrosion, decontaminates nuclear reactors and takes part in biological processes as a carrier of electrons and medical bioimaging [29, 30].”
Materials and Methods
- Page 3, 4 Lines no 90, 95, 102, 107, 112, 117 and 123, headings from 2.1 to 2.7 are not symmetrical to the margin as well as to one another and their spacing with heading numbering is not equal.
- Page 3 Line no 118, “Nuclear magnetic resonance spectra of oligomerization products were recorded with a Brucker Avane III 500 spectrometer.” may be written as “Nuclear magnetic resonance spectra of oligomerization products were recorded with a Brucker Avance III 500 spectrometer.”
- Page 4 Line no 127, “In the next step the following reagents were added: 3 mL of MMAO‑12 (modified methylaluminoxane, 7% aluminium in toluene) and 3 mL of 2‑propen‑1‑ol.” may be written as “In the next step, the following reagents were added: 3 mL of MMAO‑12 (modified methylaluminoxane, 7% aluminium in toluene) and 3 mL of 2‑propen‑1‑ol.”
- Page 4 Line no 130, “After 90 minutes of oligomerization, a white gel was obtained and then washed with a mixture of 1 M hydrochloric acid and 1 M methanol in a 1:1 molar ratio.” may be written as “After 90 minutes of oligomerization, a white gel was obtained and then it was washed with a mixture of 1 M hydrochloric acid and 1 M methanol in a 1:1 molar ratio.”
Results and discussion
- Heading of results and discussion is not symmetrical to that of introduction and material and methods.
- Page 4 Line no 141, “After comparing the results in the table above, we can conclude that the synthesis of the compound was correct because the literature values and the values of elemental analysis are similar.” may be written as “After comparing the results in the table above, we can conclude that the synthesis of the compound was correct because the literature values and the values of elemental analysis were”
- Page 4 Line no 146, “Since this energy is quantized, only radiation with certain energies specific to the functional groups performing the vibrations is absorbed.” may be written as “Since this energy is quantized, only radiation with certain energies, specific to the functional groups performing the vibrations, is absorbed.”
- Page 4 Line no 150, “It has been proven that with IR spectroscopy, hydrogen bonding, protonation and chain propagation can be investigated, provided that aquization is used fast enough.“ may be written as “It has been proven that with IR spectroscopy, hydrogen bonding, protonation and chain propagation can be investigated, provided that aquization(what is this word?) used is fast enough.“
- Page 4 Line no 152, “. The results of the IR studies show that the end product of the oligomerization contains a double bond located at one end of the chain.” may be written as “. The results of the IR studies showed that the end product of the oligomerization contained a double bond located at one end of the chain.”
- Page 5 Line no 168, “The appropriate units were assigned to the peaks in the spectra formed in the process of 2 propen 1 ol oligomerization catalyzed by [VO(dipic)(H2O)2] · 2 H2O.” may be written as “The appropriate units were assigned to the peaks in the spectra formed in the process of 2-propen-1-ol oligomerization catalyzed by [VO(dipic)(H2O)2] · 2 H2O.”
- Page 6 Line no 170, “The 649.999 m/z peak was derived from 2,5‑dihydroxybenzoic acid – matrix and molecular peak was identified with a mass/charge ratio of 703.937 m/z which contained 2 propen 1 ol 12 units.” may be written as “The 649.999 m/z peak was derived from 2,5‑dihydroxybenzoic acid – matrix and molecular peak was identified with a mass/charge ratio of 703.937 m/z which contained 2-propen-1-ol 12 units.”
- Page 6 Line no 173, “This is a confirmation that the obtained 2‑propen‑1‑ol mixture the oligomers contained chains consisting of 12, 15 and 18 allyl alcohol units.” may be written as “This was a confirmation that in the obtained 2‑propen‑1‑ol mixture, the oligomers contained chains consisting of 12, 15 and 18 allyl alcohol units.”
- Page 8 Line no 221, Formula of catalytic activity is not cleared.
- Page 8 Line no 227, “The literature values have been compared in Table 5 with the result of the calculations to find out how effective the precatalyst is.” may be written as “The literature values were compared in Table 5 with the result of the calculations to find out how effective the precatalyst was.”
- Page 9 Line no 231, “Comparing the calculatedresults with the literature values, we concluded that the precatalyst [VO(dipic)(H2O)2] · 2 H2O belonged to the group of catalysts with high catalytic activity.” may be written as “Comparing the calculated results with the literature values, we concluded that the precatalyst [VO(dipic)(H2O)2] · 2 H2O belonged to the group of catalysts with high catalytic activity.”
- Page 9 Line no 233, “The highest catalytic activity in the research to date has been noticed with olefins containing chlorine in their structure for example 2‑chloro‑2‑propen‑1‑ol.” may be written as “The highest catalytic activity in the research to date had been noticed with olefins containing chlorine in their structure for example 2‑chloro‑2‑propen‑1‑ol.”
- Page 9 Line no 235, “Thus achieving high purity and process efficiency could be increased.” may be written as “Thus by achieving high purity, process efficiency could be increased./ Thus, high purity and process efficiency can be increased.”
- Page 9 Line no 235, “The use of ligands also plays an important role.” may be written as “The use of ligands also played an important role.”
- Page 9 Line no 236, “Too extensive chains of organic clusters cause spherical hindrance, and thus low selectivity.” may be written as “Too extensive chains of organic clusters caused spherical hindrance, and thus low selectivity.”
Mechanism of the oligomerization reaction of 2-propen-1-ol catalyzed by 239 [VO(dipic)(H2O)2] •2H2O + MMAO-12
- The whole mechanism should be in past tense.
- Page 9 Line no 242, “The mechanism of the oligomerization reaction of 2‑propen‑1‑ol catalyzed by [VO(dipic)(H2O)2] · 2 H2O, with the participation of an activator (MMAO‑12), follows the mechanism of coordination polymerization (Fig. 4).” may be written as “The mechanism of the oligomerization reaction of 2‑propen‑1‑ol catalyzed by [VO(dipic)(H2O)2] · 2 H2O, with the participation of an activator (MMAO‑12), followed the mechanism of coordination polymerization (Fig. 4).”
- Page 9 Line no 244, “The approach of allyl alcohol to the center of oxovanadium(IV) with the participation of MMAO‑12 causes the formation of π complex between the alcohol's double bond and the active center (Step 1).” may be written as “The approach of allyl alcohol to the center of oxovanadium(IV) with the participation of MMAO‑12 caused the formation of π complex between the alcohol's double bond and the active center (Step 1).”
- Page 9 Line no 246, “In the next step, coupling takes place between the terminal carbon atom and the active center of oxovanadium(IV), at the expense of the water molecule from the precatalyst, which migrates to the activator.” may be written as “In the next step, coupling took place between the terminal carbon atom and the active center of oxovanadium(IV), at the expense of the water molecule from the precatalyst, which migrated to the activator.”
- Page 9 Line no 249, “The resulting electrophilic center – carbocation, then undergoes a nucleophilic attack by the π bond of allylic alcohol thus propagating alkyl chain (Step 2).” may be written as “The resulting electrophilic center – carbocation, then undergo a nucleophilic attack by the π bond of allylic alcohol thus propagated alkyl chain (Step 2).”
- Page 9 Line no 251, “In the elimination stage, the obtained oligomer separates from the dipicolinate complex of oxovanadium(IV) and modifies methylaluminoxane, through the migration of a water molecule from MMAO‑12 to the active center (Step 3).” may be written as “In the elimination stage, the obtained oligomer separated from the dipicolinate complex of oxovanadium(IV) and modified methylaluminoxane, through the migration of a water molecule from MMAO‑12 to the active center (Step 3).”
- Page 9 Line no 253, “. This transfer leads to the reconstruction of the post‑metallocene catalyst. notation.” may be written as “. This transfer led to the reconstruction of the post‑metallocene catalyst. Notation.”
- Page 9 Line no 254, “Washing the oligomer with a mixture of diluted hydrochloric acid and methanol removes residual catalyst and activator.” may be written as “Washing the oligomer with a mixture of diluted hydrochloric acid and methanol removed residual catalyst and activator.”
- Page 9 Line no 256, “The washing step is important because it The activator also reverts to its native form and the cleaved oligomer, due to the momentary and opposite polarity on the carbon atoms, can produce a polymeric structure written according to the standard can restore the stable structure of the catalyst under hydrolytic conditions.” may be written as “The washing step was important because the activator also reverted to its native form and the cleaved oligomer, due to the momentary and opposite polarity on the carbon atoms, could produce a polymeric structure written according to the standard could restore the stable structure of the catalyst under hydrolytic conditions.”
- Page 9 Line no 259, “s. Another fact confirming the correctness of the proposed termination step, involved the experiment, in which we added water to the reaction mixture after the oligomerization process.” may be written as “s. Another fact confirmed the correctness of the proposed termination step, involved the experiment, in which we added water to the reaction mixture after the oligomerization process.”
- Page 9 Line no 263, “It is worth emphasizing that this mechanism is also based on the fact that the greater the weight of a polymer, the better properties as a leaving group it constitutes thus making this reaction self‑limiting.” may be written as “It was worth emphasizing that this mechanism was also based on the fact that the greater the weight of a polymer, the better properties as a leaving group it constituted thus making this reaction self‑limiting.”
Conclusions
- Page 11 Line no 278, “It can be stated that the dipicolinate complex of oxovanadium(IV) is a highly active precatalyst with 191.53 g ·mmol‑1 · bar‑1 · h‑1 catalytic activity value.” may be written as “It could be stated that the dipicolinate complex of oxovanadium(IV) was a highly active precatalyst with 191.53 g ·mmol‑1 · bar‑1 · h‑1 catalytic activity value.”
- Page 11 Line no 80, “. Our results show that further examinations towards the potential olefin oligomerization precatalysts need to be proceeded, especially those derived from dipocolinate oxovanadium(IV) complex.” may be written as “. Our results showed that further examinations towards the potential olefin oligomerization precatalysts need to be proceeded, especially those derived from dipocolinate oxovanadium(IV) complex.”
Tables
- Page 4 Table 1 and Page 5 Table 2 have unequal spacing.
- Size as well as line spacing of Table 3 and 4 is not equal.
- Page 8 Table 5 heading is not symmetrical to that of other tables.
Figures
- Figure 1, 2 and 3 are not symmetrical to the one another.
- Fig 2 needs readjustment (going out of the margin of the paper).
References
- Page 13 Line no 348-349 and page 14 Line no 369, 376, 377 Tab spacing is different from that of other references.
Questions
The explanation of results of IR studies is not properly explained (Given only in last line). General information is given more but not that related to the study. It needs more explanation. How results of IR studies are related to the process performed?
Author Response
Answers to the Reviewers’ comments
We are very grateful to the Reviewers for their time and constructive comments on our manuscript. We have implemented their comments and suggestions and wish to submit a revised version of the manuscript for further consideration in the Journal. Changes in the initial version of the manuscript are highlighted in yellow in the revised version. Below, we also provide a point-by-point response explaining how we have addressed each of the Reviewers’ comment.
Answers to the Reviewer #1:
Comment:
Page 1 Line no 27, “Thus, the aim of our research was the examination of catalytic activity the oxovanadium(IV) dipicolinate complex [VO(dipic)(H2O)2].” may be written as “Thus, the aim of our research was the examination of catalytic activity of the oxovanadium(IV) dipicolinate complex [VO(dipic)(H2O)2].”
Authors' Response:
According to the Reviewer‘s recommendation we changed the sentence constructions in the revised version of the manuscript as follows:
“Thus, the aim of our research was the examination of catalytic activity of the oxovanadium(IV) dipicolinate complex [VO(dipic)(H2O)2] · 2 H2O (dipic = pyridine-2,6-dicarboxylate anion) in 2-propen-1-ol oligomerization.”
Comment:
Page 1 Line no 32, “The oligomerization process took place at room temperature, under atmospheric pressure and under nitrogen atmosphere to prevent oxidation of the activator MMAO‑12.” The full form of MMAO-12 should be mentioned in the abstract.
Authors' Response:
According to the Reviewer‘s recommendation the appropriate changes have been made:
“…of the activator MMAO-12 - the modified methylaluminoxane (7 wt.%) aluminum in toluene.”
Comment:
Page 2 Line no 42, “When you hear "polymer", the first thing comes to your mind are plastics?” may be written as” When you hear "polymer", the first thing comes to your mind is plastics?”
Authors' Response:
According to the Reviewer‘s recommendation we changed the sentence constructions in the revised version of the manuscript:
“When you hear "polymer", the first thing comes to your mind is plastics?”
Comment:
Page 2 Line no 49, “It has become popular to use metallocenes, i.e complex compounds containing d‑block metals and organic ligands (precatalyst) [20, 21].” may be written as “It has become popular to use metallocenes, e. complex compounds containing d‑block metals and organic ligands (precatalyst) [20, 21]."
Authors' Response:
According to the Reviewer‘s recommendation the appropriate changes have been made:
“It has become popular to use metallocenes, e. complex compounds containing d‑block metals and organic ligands (precatalyst).”
Comment:
Page 2 Line no 56, “Modified methyl aluminoxane, as an activatoroxidizes very quickly when there is oxygen in the reaction system, therefore nitrogen is introduced to prevent this.” may be written as “Modified methyl aluminoxane, as an activator, oxidizes very quickly when there is oxygen in the reaction system therefore, nitrogen is introduced to prevent this. “
Authors' Response:
According to the Reviewer‘s recommendation we changed the sentence constructions in the revised version of the manuscript as follows:
“Modified methyl aluminoxane, as an activator, oxidizes very quickly when there is oxygen in the reaction system therefore, nitrogen is introduced to prevent this.”
Comment:
Page 2 Line no 59, “The subject of interest of scientists at the beginning of the 20th century was the comparison of the trans spatial structure [VO(dipic)(H2O)2] · 2 H2O (1) to the known compound [VO(dipic)(o‑phen)] · 3 H2O (2) by X‑ray crystallography and obtaining (2) from (1) by substituting two water molecules with 1,10‑phenanthroline.” may be written as “The subject of interest of scientists at the beginning of the 20th century was the comparison of the trans spatial structure [VO(dipic)(H2O)2] · 2 H2O (1) to the known compound [VO(dipic)(o‑phen)] · 3 H2O (2) by X‑ray crystallography and to obtain/obtained (2) from (1) by substituting two water molecules with 1,10‑phenanthroline.”
Authors' Response:
According to the Reviewer‘s recommendation the appropriate changes have been made:
„The subject of interest of scientists at the beginning of the 20th century was the comparison of the trans spatial structure [VO(dipic)(H2O)2] · 2 H2O (1) to the known compound [VO(dipic)(o-phen)] · 3 H2O (2) by X-ray crystallography and to obtain (2) from (1) by substituting two water molecules with 1,10-phenanthroline.
Comment:
Page 2 Line no 62, “It turned out that the coordination sphere around the oxovanadium(IV) ion was completely transformed during the reaction, which influences the kinetic aspect [28].” may be written as “It turned out that the coordination sphere around the oxovanadium(IV) ion was completely transformed during the reaction, which influenced the kinetic aspect [28].”
Authors' Response:
According to the Reviewer‘s recommendation we changed the sentence constructions in the revised version of the manuscript as follows:
“It turned out that the coordination sphere around the oxovanadium(IV) ion was completely transformed during the reaction, which influenced the kinetic aspect [28].”
Comment:
Page 2 Line no 68, “It is used to remove corrosion, decontaminates nuclear reactors, and takes part in biological processes as a carrier of electrons and medical bioimaging [29, 30].” may be written as “It is used to remove corrosion, decontaminates nuclear reactors and takes part in biological processes as a carrier of electrons and medical bioimaging [29, 30].”
Authors' Response:
According to the Reviewer‘s recommendation the appropriate changes have been made:
“It is used to remove corrosion, decontaminates nuclear reactors and takes part in biological processes as a carrier of electrons and medical bioimaging”
Comment:
Materials and Methods
Page 3, 4 Lines no 90, 95, 102, 107, 112, 117 and 123, headings from 2.1 to 2.7 are not symmetrical to the margin as well as to one another and their spacing with heading numbering is not equal.”
Authors' Response:
According to the Reviewer‘s recommendation editorial errors have been corrected.
Comment:
Page 3 Line no 118, “Nuclear magnetic resonance spectra of oligomerization products were recorded with a Brucker Avane III 500 spectrometer.” may be written as “Nuclear magnetic resonance spectra of oligomerization products were recorded with a Brucker Avance III 500 spectrometer.”
Authors' Response:
According to the Reviewer‘s recommendation the appropriate changes have been made:
“Nuclear magnetic resonance spectra of oligomerization products were recorded with a Brucker Avance III 500 spectrometer.”
Comment:
Page 4 Line no 127, “In the next step the following reagents were added: 3 mL of MMAO‑12 (modified methylaluminoxane, 7% aluminium in toluene) and 3 mL of 2‑propen‑1‑ol.” may be written as “In the next step, the following reagents were added: 3 mL of MMAO‑12 (modified methylaluminoxane, 7% aluminium in toluene) and 3 mL of 2‑propen‑1‑ol.”
Authors' Response:
According to the Reviewer‘s recommendation we changed the sentence constructions in the revised version of the manuscript:
“In the next step, the following reagents were added: 3 mL of MMAO‑12 (modified methylaluminoxane, 7% aluminium in toluene) and 3 mL of 2‑propen‑1‑ol.”
Comment:
Page 4 Line no 130, “After 90 minutes of oligomerization, a white gel was obtained and then washed with a mixture of 1 M hydrochloric acid and 1 M methanol in a 1:1 molar ratio.” may be written as “After 90 minutes of oligomerization, a white gel was obtained and then it was washed with a mixture of 1 M hydrochloric acid and 1 M methanol in a 1:1 molar ratio.”
Authors' Response:
According to the Reviewer‘s recommendation the indicated part has been corrected as follows:
“After 90 minutes of oligomerization, a white gel was obtained and then it was washed with a mixture of 1 M hydrochloric acid and 1 M methanol in a 1:1 molar ratio“.
Comment:
Heading of results and discussion is not symmetrical to that of introduction and material and methods.
Authors' Response:
According to the Reviewer‘s recommendation editorial errors have been corrected.
Comment:
Page 4 Line no 141, “After comparing the results in the table above, we can conclude that the synthesis of the compound was correct because the literature values and the values of elemental analysis are similar.” may be written as “After comparing the results in the table above, we can conclude that the synthesis of the compound was correct because the literature values and the values of elemental analysis were”
Authors' Response:
According to the Reviewer‘s recommendation the appropriate changes have been made:
“After comparing the results in the table above, we can conclude that the synthesis of the compound was correct because the literature values and the values of elemental analysis were similar”
Comment:
Page 4 Line no 146, “Since this energy is quantized, only radiation with certain energies specific to the functional groups performing the vibrations is absorbed.” may be written as “Since this energy is quantized, only radiation with certain energies, specific to the functional groups performing the vibrations, is absorbed.”
Authors' Response:
According to the Reviewer‘s recommendation we changed the sentence constructions in the revised version of the manuscript:
‘’Since this energy is quantized, only radiation with certain energies, specific to the functional groups performing the vibrations, is absorbed.”
Comment:
Page 4 Line no 150, “It has been proven that with IR spectroscopy, hydrogen bonding, protonation and chain propagation can be investigated, provided that aquization is used fast enough.“ may be written as “It has been proven that with IR spectroscopy, hydrogen bonding, protonation and chain propagation can be investigated, provided that aquization(what is this word?) used is fast enough.“
Authors' Response:
According to the Reviewer‘s recommendation the appropriate changes have been made:
“It has been proven that with IR spectroscopy, hydrogen bonding, protonation and chain propagation can be investigated“.
Comment:
Page 4 Line no 152, “. The results of the IR studies show that the end product of the oligomerization contains a double bond located at one end of the chain.” may be written as “. The results of the IR studies showed that the end product of the oligomerization contained a double bond located at one end of the chain.”
Authors' Response:
According to the Reviewer‘s recommendation the appropriate changes have been made:
“The results of the IR studies showed that the end product of the oligomerization contained a double bond located at one end of the chain.”
Comment:
Page 5 Line no 168, “The appropriate units were assigned to the peaks in the spectra formed in the process of 2 propen 1 ol oligomerization catalyzed by [VO(dipic)(H2O)2] · 2 H2O.” may be written as “The appropriate units were assigned to the peaks in the spectra formed in the process of 2-propen-1-ol oligomerization catalyzed by [VO(dipic)(H2O)2] · 2 H2O.”
Authors' Response:
According to the Reviewer‘s recommendation the appropriate changes have been made:
“The appropriate units were assigned to the peaks in the spectra formed in the process of 2-propen-1-ol oligomerization catalyzed by [VO(dipic)(H2O)2] · 2 H2O. “
Comment:
Page 6 Line no 170, “The 649.999 m/z peak was derived from 2,5‑dihydroxybenzoic acid – matrix and molecular peak was identified with a mass/charge ratio of 703.937 m/z which contained 2 propen 1 ol 12 units.” may be written as “The 649.999 m/z peak was derived from 2,5‑dihydroxybenzoic acid – matrix and molecular peak was identified with a mass/charge ratio of 703.937 m/z which contained 2-propen-1-ol 12 units.”
Authors' Response:
According to the Reviewer‘s recommendation we changed the sentence constructions in the revised version of the manuscript:
’’The 649.999 m/z peak was derived from 2,5‑dihydroxybenzoic acid – matrix and molecular peak was identified with a mass/charge ratio of 703.937 m/z which contained 2-propen-1-ol 12 units.”
Comment:
Page 6 Line no 173, “This is a confirmation that the obtained 2‑propen‑1‑ol mixture the oligomers contained chains consisting of 12, 15 and 18 allyl alcohol units.” may be written as “This was a confirmation that in the obtained 2‑propen‑1‑ol mixture, the oligomers contained chains consisting of 12, 15 and 18 allyl alcohol units.”
Authors' Response:
According to the Reviewer‘s recommendation the appropriate changes have been made:
’’This was a confirmation that in the obtained 2‑propen‑1‑ol mixture, the oligomers contained chains consisting of 12, 15 and 18 allyl alcohol units.”
Comment:
Page no 5 Line no 156, “peaks on the spectra have been assigned to the appropriate units spectra formed in the process of 2‑propen‑1‑ol oligomerization catalyzed by [VO(dipic)(H2O)2] · 2 H2O. The 649.999 m/z peak is derived from 2,5‑dihydroxybenzoic acid – matrix and molecular peak was identified with a mass/charge ratio of 703.937 m/z which contains twelve 2‑propen‑1‑ol units” may be written as “The appropriate units were assigned to the peaks in the spectra formed in the process of 2‑propen‑1‑ol oligomerization catalyzed by [VO(dipic)(H2O)2] · 2 H2O. The 649.999 m/z peak was (line no. 158, page 5) derived from 2,5‑dihydroxybenzoic acid – matrix and molecular peak was identified with a mass/charge ratio of 703.937 m/z which contained 2‑propen‑1‑ol 12 units.”
Authors' Response:
According to the Reviewer‘s recommendation we changed the sentence constructions in the revised version of the manuscript:
„The appropriate units were assigned to the peaks in the spectra formed in the process of 2‑propen‑1‑ol oligomerization catalyzed by [VO(dipic)(H2O)2] · 2 H2O. The 649.999 m/z peak was derived from 2,5‑dihydroxybenzoic acid – matrix and molecular peak was identified with a mass/charge ratio of 703.937 m/z which contained 2‑propen‑1‑ol 12 units.”
Comment:
Page 8 Line no 221, Formula of catalytic activity is not cleared.
Authors' Response:
According to the Reviewer‘s recommendation the formula of catalytic activity has been explained.
Comment:
Page 8 Line no 227, “The literature values have been compared in Table 5 with the result of the calculations to find out how effective the precatalyst is.” may be written as “The literature values were compared in Table 5 with the result of the calculations to find out how effective the precatalyst was.”
Authors' Response:
According to the Reviewer‘s recommendation the appropriate changes have been made:
“The literature values were compared in Table 5 with the result of the calculations to find out how effective the precatalyst was. “
Comment:
Page 9 Line no 231, “Comparing the calculatedresults with the literature values, we concluded that the precatalyst [VO(dipic)(H2O)2] · 2 H2O belonged to the group of catalysts with high catalytic activity.” may be written as “Comparing the calculated results with the literature values, we concluded that the precatalyst [VO(dipic)(H2O)2] · 2 H2O belonged to the group of catalysts with high catalytic activity.”
Authors' Response:
According to the Reviewer‘s recommendation the appropriate changes have been made:
’’Comparing the calculated results with the literature values, we concluded that the precatalyst [VO(dipic)(H2O)2] · 2 H2O belonged to the group of catalysts with high catalytic activity.”
Comment:
Page 9 Line no 233, “The highest catalytic activity in the research to date has been noticed with olefins containing chlorine in their structure for example 2‑chloro‑2‑propen‑1‑ol.” may be written as “The highest catalytic activity in the research to date had been noticed with olefins containing chlorine in their structure for example 2‑chloro‑2‑propen‑1‑ol.”
Authors' Response:
According to the Reviewer‘s recommendation the appropriate changes have been made:
„The highest catalytic activity in the research to date had been noticed with olefins containing chlorine in their structure for example 2‑chloro‑2‑propen‑1‑ol.”
Comment:
Page 9 Line no 235, “Thus achieving high purity and process efficiency could be increased.” may be written as “Thus by achieving high purity, process efficiency could be increased./ Thus, high purity and process efficiency can be increased.”
Authors' Response:
According to the Reviewer‘s recommendation the appropriate changes have been made:
“Thus by achieving high purity, process efficiency could be increased.“
Comment:
Page 9 Line no 235, “The use of ligands also plays an important role.” may be written as “The use of ligands also played an important role.”
Authors' Response:
According to the Reviewer‘s recommendation the appropriate changes have been made:
“The use of ligands also played an important role.“
Comment:
Page 9 Line no 236, “Too extensive chains of organic clusters cause spherical hindrance, and thus low selectivity.” may be written as “Too extensive chains of organic clusters caused spherical hindrance, and thus low selectivity.”
Authors' Response:
According to the Reviewer‘s recommendation the appropriate changes have been made:
“Too extensive chains of organic clusters caused spherical hindrance, and thus low selectivity. “
Comment:
The whole mechanism should be in past tense. Page 9 Line no 242, “The mechanism of the oligomerization reaction of 2‑propen‑1‑ol catalyzed by [VO(dipic)(H2O)2] · 2 H2O, with the participation of an activator (MMAO‑12), follows the mechanism of coordination polymerization (Fig. 4).” may be written as “The mechanism of the oligomerization reaction of 2‑propen‑1‑ol catalyzed by [VO(dipic)(H2O)2] · 2 H2O, with the participation of an activator (MMAO‑12), followed the mechanism of coordination polymerization (Fig. 4).”
Authors' Response:
According to the Reviewer‘s recommendation the appropriate changes have been made as follows:
“The mechanism of the oligomerization reaction of 2‑propen‑1‑ol catalyzed by [VO(dipic)(H2O)2] · 2 H2O, with the participation of an activator (MMAO‑12), followed the mechanism of coordination polymerization (Fig. 4). “
Comment:
Page 9 Line no 244, “The approach of allyl alcohol to the center of oxovanadium(IV) with the participation of MMAO‑12 causes the formation of π complex between the alcohol's double bond and the active center (Step 1).” may be written as “The approach of allyl alcohol to the center of oxovanadium(IV) with the participation of MMAO‑12 caused the formation of π complex between the alcohol's double bond and the active center (Step 1).”
Authors' Response:
According to the Reviewer‘s recommendation the appropriate changes have been made:
“The approach of allyl alcohol to the center of oxovanadium(IV) with the participation of MMAO‑12 caused the formation of π complex between the alcohol's double bond and the active center (Step 1).“
Comment:
Page 9 Line no 246, “In the next step, coupling takes place between the terminal carbon atom and the active center of oxovanadium(IV), at the expense of the water molecule from the precatalyst, which migrates to the activator.” may be written as “In the next step, coupling took place between the terminal carbon atom and the active center of oxovanadium(IV), at the expense of the water molecule from the precatalyst, which migrated to the activator.”.
Authors' Response:
According to the Reviewer‘s recommendation the appropriate changes have been made:
“In the next step, coupling took place between the terminal carbon atom and the active center of oxovanadium(IV), at the expense of the water molecule from the precatalyst, which migrated to the activator. “
Comment:
Page 9 Line no 249, “The resulting electrophilic center – carbocation, then undergoes a nucleophilic attack by the π bond of allylic alcohol thus propagating alkyl chain (Step 2).” may be written as “The resulting electrophilic center – carbocation, then undergo a nucleophilic attack by the π bond of allylic alcohol thus propagated alkyl chain (Step 2).”
Authors' Response:
According to the Reviewer‘s recommendation the appropriate changes have been made as follows:
“The resulting electrophilic center – carbocation, then undergo a nucleophilic attack by the π bond of allylic alcohol thus propagated alkyl chain (Step 2). “
Comment:
Page 9 Line no 251, “In the elimination stage, the obtained oligomer separates from the dipicolinate complex of oxovanadium(IV) and modifies methylaluminoxane, through the migration of a water molecule from MMAO‑12 to the active center (Step 3).” may be written as “In the elimination stage, the obtained oligomer separated from the dipicolinate complex of oxovanadium(IV) and modified methylaluminoxane, through the migration of a water molecule from MMAO‑12 to the active center (Step 3).”
Authors' Response:
According to the Reviewer‘s recommendation the appropriate changes have been made:
“In the elimination stage, the obtained oligomer separated from the dipicolinate complex of oxovanadium(IV) and modified methylaluminoxane, through the migration of a water molecule from MMAO‑12 to the active center (Step 3).”
Comment:
Page 9 Line no 254, “Washing the oligomer with a mixture of diluted hydrochloric acid and methanol removes residual catalyst and activator.” may be written as “Washing the oligomer with a mixture of diluted hydrochloric acid and methanol removed residual catalyst and activator.”
Authors' Response:
According to the Reviewer‘s recommendation the appropriate changes have been made:
“Washing the oligomer with a mixture of diluted hydrochloric acid and methanol removed residual catalyst and activator. “
Comment:
Page 9 Line no 256, “The washing step is important because it The activator also reverts to its native form and the cleaved oligomer, due to the momentary and opposite polarity on the carbon atoms, can produce a polymeric structure written according to the standard can restore the stable structure of the catalyst under hydrolytic conditions.” may be written as “The washing step was important because the activator also reverted to its native form and the cleaved oligomer, due to the momentary and opposite polarity on the carbon atoms, could produce a polymeric structure written according to the standard could restore the stable structure of the catalyst under hydrolytic conditions.”
Authors' Response:
According to the Reviewer‘s recommendation the appropriate changes have been made as follows:
“The washing step was important because the activator also reverted to its native form and the cleaved oligomer, due to the momentary and opposite polarity on the carbon atoms, could produce a polymeric structure written according to the standard could restore the stable structure of the catalyst under hydrolytic conditions.“
Comment:
Page 9 Line no 259, “s. Another fact confirming the correctness of the proposed termination step, involved the experiment, in which we added water to the reaction mixture after the oligomerization process.” may be written as “s. Another fact confirmed the correctness of the proposed termination step, involved the experiment, in which we added water to the reaction mixture after the oligomerization process.”
Authors' Response:
According to the Reviewer‘s recommendation the appropriate changes have been made:
“Another fact confirmed the correctness of the proposed termination step, involved the experiment, in which we added water to the reaction mixture after the oligomerization process. “
Comment:
Page 9 Line no 263, “It is worth emphasizing that this mechanism is also based on the fact that the greater the weight of a polymer, the better properties as a leaving group it constitutes thus making this reaction self‑limiting.” may be written as “It was worth emphasizing that this mechanism was also based on the fact that the greater the weight of a polymer, the better properties as a leaving group it constituted thus making this reaction self‑limiting.”
Authors' Response:
According to the Reviewer‘s recommendation the appropriate changes have been made as follows:
“It was worth emphasizing that this mechanism was also based on the fact that the greater the weight of a polymer, the better properties as a leaving group it constituted thus making this reaction self‑limiting. “
Comment:
Page 11 Line no 278, “It can be stated that the dipicolinate complex of oxovanadium(IV) is a highly active precatalyst with 191.53 g · mmol‑1 · bar‑1 · h‑1 catalytic activity value.” may be written as “It could be stated that the dipicolinate complex of oxovanadium(IV) was a highly active precatalyst with 191.53 g · mmol‑1 · bar‑1 · h‑1 catalytic activity value.”
Authors' Response:
According to the Reviewer‘s recommendation the appropriate changes have been made as follows:
“It could be stated that the dipicolinate complex of oxovanadium(IV) was a highly active precatalyst with 191.53 g · mmol‑1 · bar‑1 · h‑1 catalytic activity value. “
Comment:
Page 11 Line no 80, “. Our results show that further examinations towards the potential olefin oligomerization precatalysts need to be proceeded, especially those derived from dipocolinate oxovanadium(IV) complex.” may be written as “. Our results showed that further examinations towards the potential olefin oligomerization precatalysts need to be proceeded, especially those derived from dipocolinate oxovanadium(IV) complex.”
Authors' Response:
According to the Reviewer‘s recommendation the appropriate changes have been made as follows:
“Our results showed that further examinations towards the potential olefin oligomerization precatalysts need to be proceeded, especially those derived from dipocolinate oxovanadium(IV) complex. “
Comment:
Page 4 Table 1 and Page 5 Table 2 have unequal spacing.
Size as well as line spacing of Table 3 and 4 is not equal.
Page 8 Table 5 heading is not symmetrical to that of other tables.
Figure 1, 2 and 3 are not symmetrical to the one another.
Fig 2 needs readjustment (going out of the margin of the paper).
Page 13 Line no 348-349 and page 14 Line no 369, 376, 377 Tab spacing is different from that of other references.
Authors' Response:
Editorial errors have been corrected.
Comment:
The explanation of results of IR studies is not properly explained (Given only in last line). General information is given more but not that related to the study. It needs more explanation. How results of IR studies are related to the process performed?
Authors' Response:
The IR studies confirmed the structure of the oligomerization products. It shows that the correct products were obtained.

Reviewer 2 Report
I think it is very meaningful to be able to make polymers/oligomers through the new catalyst system.
But it looks like you need to check a few things.
1. Currently, the author proceeded to manufacture an oligomer using vinyl alcohol. In this regard, has the extensibility to other materials been confirmed?
2. Is there any reason why the repeat number of vinyl alcohol oligomers is determined to be 12/15/18? If the reaction continues, will it continue to grow as a multiple of 3, such as 21/24?
Author Response
Answers to the Reviewers’ comments
We are very grateful to the Reviewers for their time and constructive comments on our manuscript. We have implemented their comments and suggestions and wish to submit a revised version of the manuscript for further consideration in the Journal. Changes in the initial version of the manuscript are highlighted in yellow in the revised version. Below, we also provide a point-by-point response explaining how we have addressed each of the Reviewers’ comment.
Answers to the Reviewer #2:
Comment:
I think it is very meaningful to be able to make polymers/oligomers through the new catalyst system.
Authors' Response:
Thank you very much.
Comment:
Currently, the author proceeded to manufacture an oligomer using vinyl alcohol. In this regard, has the extensibility to other materials been confirmed?
Authors' Response:
Thank you very much for these valuable comments. In the future, we plan to carry out oligomerization and polymerization of other olefins, such as ethylene, propylene with dipicolinate oxovanadium(IV) complex compound as a precatalyst.
Changes in the initial version of the manuscript are highlighted in yellow in the revised version:
„Thorn et al. have reported the application of V-dipic complexes as: [VO(dipic)(i-PrO)], [VO(dipic)(pinme)], [VO(dipic)(dpheol)] and analogs in stoichiometric aerobic oxidation of isopropanol and other alcohols as lignin models [38]. Another example is Gawdzik et al. reported new oxovanadium(IV) microclusters with 2-phenylpyridine which has shown highly activity for the 3-buten-1-ol, 2-chloro-2-propen-1-ol, allyl alcohol, and 2,3-dibromo-2-propen-1-ol oligomerizations [39].“
Comment:
Is there any reason why the repeat number of vinyl alcohol oligomers is determined to be 12/15/18? If the reaction continues, will it continue to grow as a multiple of 3, such as 21/24?
Authors' Response:
There is no element of the repetition of the formation of the mers. Depending on how many mers are present in the MALDI-TOF-MS spectrum, we can determine whether the catalyst leads to the formation of an oligomer or a polymer. It is impossible to predict how much the mer chain will have if, for example, a polymerization activity is performed.

Reviewer 3 Report
The manuscript describes the oligomerization of 2-propen-1-ol with a homogeneous oxovanadium(IV) dipicolinate complex using toluene as solvent under ambient conditions (room temperature and ambient pressure) and nitrogen atmosphere. The authors present IR data to characterize the synthesized vanadium complex and MALDI-TOF-MS, IR, and NMR characterization of the oligomeric products produced after the reaction. In the end, the authors present a reaction mechanism that involves initiation, chain propagation, and termination steps.
Overall, the quality of the English should be improved, and the paper contains sentences with typos or out of formatting. The introduction is too short and does not present previous relevant literature on using homogeneous catalysts for 2-propen-1-ol oligomerization. In my opinion, the authors should discuss the relevance of this process and the applications for 2-propen-1-ol oligomers. The experimental section lacks sufficient details to allow other researchers in the field to replicate the experiments described in the manuscript. Also, further characterization of the synthesized vanadium complex should be performed to validate its chemical structure since the level of detail presented in Figure 4 for the chemical nature of the vanadium complex is not consistent with the experimental results described in the manuscript. The only characterization provided is the elemental analysis shown in Table 1. The characterization of oligomeric products via IR, MALDI-TOF-MS, and NMR should be better discussed since the discussion centers too much on how these techniques work instead of the actual significance of the results. Finally, the results presented throughout the manuscript are insufficient to develop such a detailed reaction mechanism, as shown in Figure 4. At best, the authors would be able to propose a reaction pathway based on the MS results. For these reasons, I believe that the manuscript is not suited for publication in its current state.
Please see the major and minor comments below:
Major comments:
1 English and formatting
The quality of the English should be improved, and some sentences contain typos or are out of formatting. Refer to some of these sentences below (typos in bold):
- Page 2, line 56: “Modified methyl aluminoxane, as an activatoroxidizes”
- Page 3, line 101: “…out on a Bruker IFS 66 spectrometr.”
- Page 9, line 231: “Comparing the calculatedresults with the literature values”
- Page 9, line 256: “The washing step is important because it The activator also reverts to its native”
- Page 9, line 253: “…of a water molecule from MMAO‑12 to the active center (Step 3),.”
- Page 9, line 254: “…the reconstruction of the post‑metallocene notation. Washing the oligomer…”
2 Experimental section
The authors should provide the temperature used to heat the solution to synthesize the dipicolinate oxovanadium complex (Page 3, line 98). Also, how long did it take to cool the solution? One month seems a long time to crystallize the dipicolinate oxovanadium crystals. What temperature was used during crystallization?
Sample preparation for the elemental analysis should be included in section 2.3 (Page 3, line 104).
The authors should include the resolution used for IR measurements and the detector used in the spectrometer.
There are too many significant digits for the NMR frequencies in section 2.6.
The authors should include a process flow diagram for the oligomerization system used in section 2.7. Also, could the authors define ordinary pressure?
3 Results and discussion
How did the authors estimate the elemental composition of the vanadium complex using theoretical calculations? The equations used and a thorough description of the method should be included in the manuscript.
The authors should include others characterization results for the vanadium complex. I suggest the authors include IR and XPS of the synthesized catalyst. These additional analyses would help validate the structure of the vanadium complex and its biding environment/oxidation state.
The authors should include the actual IR spectrum of the products in addition to the table with the vibrational frequencies. Also, the vibrational frequencies shown in Table 2 contain too many significant digits. The discussion on page 4, lines 145-153 is unnecessary since the reader should be familiar with IR characterization. The same comment goes to page 7, lines 191-203 for the NMR analysis.
Asserting the double bond location just from IR analysis seems too speculative, and additional analysis should be included to validate this hypothesis. The MS spectrum shown in Figure 1 suggests different products with the same m/z value, indicating isomers among the oligomeric products. These isomers can have internal double bonds or branches. The authors should include not only the m/z but also the match with some potential components relative to the NIST library. This data would help develop a reaction pathway consistent with the data presented in the manuscript.
What is the uncertainty of the catalytic activity calculation? This measure would be important to provide a precise comparison with the previous values reported in the literature – catalytic activities shown in Table 5. I also suggest that the authors show data for the consumption of 2-propen-1-ol and provide the mass balance for the experiment. This would be important to validate the catalytic activity calculated in the manuscript. Is Va4+ the only oxidation state present in the synthesized catalyst? XPS data would also help in this regard.
The level of detail for the mechanism presented in Figure 4 does not match the level of detail of the experimental data discussed in the manuscript, and the conclusions for the initiation, propagation, and termination steps of the oligomerization processes are far too speculative. I recommend that the authors consider the MS data when developing a reaction pathway instead of a detailed reaction mechanism.
Minor comments:
- The quality of the images should be improved. Also, Figures 2 and 3 are beyond the limit of the margin.

Author Response
Answers to the Reviewers’ comments
We are very grateful to the Reviewers for their time and constructive comments on our manuscript. We have implemented their comments and suggestions and wish to submit a revised version of the manuscript for further consideration in the Journal. Changes in the initial version of the manuscript are highlighted in yellow in the revised version. Below, we also provide a point-by-point response explaining how we have addressed each of the Reviewers’ comment.
Answers to the Reviewer #3:
Comment:
The manuscript describes the oligomerization of 2-propen-1-ol with a homogeneous oxovanadium(IV) dipicolinate complex using toluene as solvent under ambient conditions (room temperature and ambient pressure) and nitrogen atmosphere. The authors present IR data to characterize the synthesized vanadium complex and MALDI-TOF-MS, IR, and NMR characterization of the oligomeric products produced after the reaction. In the end, the authors present a reaction mechanism that involves initiation, chain propagation, and termination steps.
Overall, the quality of the English should be improved, and the paper contains sentences with typos or out of formatting. The introduction is too short and does not present previous relevant literature on using homogeneous catalysts for 2-propen-1-ol oligomerization. In my opinion, the authors should discuss the relevance of this process and the applications for 2-propen-1-ol oligomers. The experimental section lacks sufficient details to allow other researchers in the field to replicate the experiments described in the manuscript. Also, further characterization of the synthesized vanadium complex should be performed to validate its chemical structure since the level of detail presented in Figure 4 for the chemical nature of the vanadium complex is not consistent with the experimental results described in the manuscript. The only characterization provided is the elemental analysis shown in Table 1. The characterization of oligomeric products via IR, MALDI-TOF-MS, and NMR should be better discussed since the discussion centers too much on how these techniques work instead of the actual significance of the results. Finally, the results presented throughout the manuscript are insufficient to develop such a detailed reaction mechanism, as shown in Figure 4. At best, the authors would be able to propose a reaction pathway based on the MS results. For these reasons, I believe that the manuscript is not suited for publication in its current state.
English and formatting
The quality of the English should be improved, and some sentences contain typos or are out of formatting. Refer to some of these sentences below (typos in bold):
- Page 2, line 56: “Modified methyl aluminoxane, as an activatoroxidizes”
- Page 3, line 101: “…out on a Bruker IFS 66 spectrometr.”
- Page 9, line 231: “Comparing the calculatedresults with the literature values”
- Page 9, line 256: “The washing step is important because it The activator also reverts to its native”
- Page 9, line 253: “…of a water molecule from MMAO‑12 to the active center (Step 3),.”
- Page 9, line 254: “…the reconstruction of the post‑metallocene notation. Washing the oligomer…”
Authors' Response:
Thank you very much for these valuable comments. The aim of our work was to investigate the catalytic properties of oxovanadium(IV) complex compound, therefore we did not focus in detail on the application of the oligomer. We sincerely apologize for the linguistic errors. All typos have been corrected. Changes in the initial version of the manuscript are highlighted in yellow in the revised version.
Comment:
The authors should provide the temperature used to heat the solution to synthesize the dipicolinate oxovanadium complex (Page 3, line 98). Also, how long did it take to cool the solution? One month seems a long time to crystallize the dipicolinate oxovanadium crystals. What temperature was used during crystallization?
Authors' Response:
According to the Reviewer‘s recommendation the appropriate changes have been made as follows:
“The entire solution was heated at 100 °C to reflux for 90 minutes until the mixture changed color. It took 30 minutes to cool down. Crystallization was carried out at room temperature.“
Comment:
Sample preparation for the elemental analysis should be included in section 2.3 (Page 3, line 104).
Authors' Response:
The samples tested by means of elemental analysis were dry, homogeneous with a mass of 2 mg. This information has been added in the revised version of the manuscript.
Comment:
The authors should include the resolution used for IR measurements and the detector used in the spectrometer.
Authors' Response:
According to the Reviewer‘s recommendation the appropriate changes have been made as follows:
“The IFS66 apparatus by BRUKER (1995) to perform infrared spectra in the Fourier tra nsform with a resoluteion of 0.12 cm-1 for solid, liquid and gaseous samples in the entire range, i.e. MIR (4000 - 400 cm-1), FIR (700 - 4.0 cm-1). DLATGS was a detector in IR measurements.“
Comment:
There are too many significant digits for the NMR frequencies in section 2.6.
Authors' Response:
According to the Reviewer‘s recommendation the appropriate changes have been made as follows:
“The measurement was carried out at 25 °C. The measured frequency was 126 MHz for 13C NMR and 500 MHz for 1H NMR.“
Comment:
The authors should include a process flow diagram for the oligomerization system used in section 2.7. Also, could the authors define ordinary pressure?
Authors' Response:
According to the Reviewer‘s recommendation a process flow diagram for the oligomerization system was added to the manuscript (Fig. 1.). The ordinary pressure was equal to 1013 hPa.
Fig. 1. A process flow diagram for the oligomerization system.
Comment:
How did the authors estimate the elemental composition of the vanadium complex using theoretical calculations? The equations used and a thorough description of the method should be included in the manuscript.
Authors' Response:
In accordance with the reviewer's comments, we have added a calculation method.
“Theoretical data have been calcuted according to the following procedure:
Mass of [VO(dipic)(H2O)2] · 2 H2O = 305.9 g/mol
%C = 84/305,9 * 100% = 27.46%
%H = 13/305.9 *100% = 4.25%
%N = 14/305.9 *100% = 4.58%”
Comment:
The authors should include others characterization results for the vanadium complex. I suggest the authors include IR and XPS of the synthesized catalyst. These additional analyses would help validate the structure of the vanadium complex and its biding environment/oxidation state.
Authors' Response:
According to the Reviewer‘s recommendation IR spectrum has been added.
“In order to confirm the structure of the synthesized crystal of complex compound, we described the IR spectrum of the dipicolinate oxovanadium(IV) complex (Fig. 2).“
Fig. 2. IR spectrum of [VO(dipic)(H2O)2] · 2 H2O.
|
Wavenumber [cm-1] |
Type of vibration with function group |
|
3571 cm−1 |
v(OH) |
|
1665 cm−1 |
v(COO) of dipic |
|
1352 cm−1 |
v(COO) of dipic |
|
983 cm−1 |
V=O stretching frequency |
|
452 cm−1 |
stretching vibration of the V-N |
Comment:
The authors should include the actual IR spectrum of the products in addition to the table with the vibrational frequencies. Also, the vibrational frequencies shown in Table 2 contain too many significant digits. The discussion on page 4, lines 145-153 is unnecessary since the reader should be familiar with IR characterization. The same comment goes to page 7, lines 191-203 for the NMR analysis.
Authors' Response:
According to the Reviewer‘s recommendation IR spectrum has been added. The numeric values in the table have been changed to not include too many significant numbers. We agree with the reviewer's opinion, however, these additional informations regarding the characteristics of spectroscopy was added due to comments from other reviewers.
Comment:
Asserting the double bond location just from IR analysis seems too speculative, and additional analysis should be included to validate this hypothesis. The MS spectrum shown in Figure 1 suggests different products with the same m/z value, indicating isomers among the oligomeric products. These isomers can have internal double bonds or branches. The authors should include not only the m/z but also the match with some potential components relative to the NIST library. This data would help develop a reaction pathway consistent with the data presented in the manuscript.
Authors' Response:
We agree with the Reviewer’s opinion. However, despite our efforts to find the spectra in the NIST library, we were unable to find them.
Comment:
What is the uncertainty of the catalytic activity calculation? This measure would be important to provide a precise comparison with the previous values reported in the literature – catalytic activities shown in Table 5. I also suggest that the authors show data for the consumption of 2-propen-1-ol and provide the mass balance for the experiment. This would be important to validate the catalytic activity calculated in the manuscript. Is V4+ the only oxidation state present in the synthesized catalyst? XPS data would also help in this regard.
Authors' Response:
Thank you for your comment. The aim of the research was not to characterize the obtained oligomer, but to investigate the catalytic properties of the catalyst in the oligomerization process. We did not perform validation due to the need to perform repeated processes, which is associated with higher costs. According to the IR analysis, we are sure that V4+ is the only oxidation state in the product. XPS analysis is not possible due to the lack of equipment in educational institutions in which we work. However, we will take into account some valuable suggestions from the editor for future research.
Comment:
The level of detail for the mechanism presented in Figure 4 does not match the level of detail of the experimental data discussed in the manuscript, and the conclusions for the initiation, propagation, and termination steps of the oligomerization processes are far too speculative. I recommend that the authors consider the MS data when developing a reaction pathway instead of a detailed reaction mechanism.
Authors' Response:
Thank you for your valuable comments. The mechanism we have created is a proposed mechanism based on literature, not research. We will put the appropriate annotation in the manuscript. We realize that in order to confirm the reaction mechanism, more spectral analyzes and additional research are needed.
Comment:
The quality of the images should be improved. Also, Figures 2 and 3 are beyond the limit of the margin.
Authors' Response:
The quality of the images have been improved. Figures have been reduced so that they do not exceed the margin.

Round 2
Reviewer 1 Report
No more revision required.
Author Response
We are very grateful to the Reviewer for their time and constructive comments on our manuscript. We have implemented their comments and suggestions and wish to submit a revised version of the manuscript for further consideration in Materials
Reviewer 2 Report
Thank you for answer
I have a few additional questions.
I have another question in terms of utilization rather than design of polymers.
1. Check the catalyst efficiency. If so, is the recovery of the catalyst no problem? In Fig 7, it appears to be completely separated, is there no such problem?
2. Is there any way to determine the exact molecular size? (Not measurement/it means Poltmerization process) If not, I think it is difficult to explain the difference from the existing catalyst system.
Author Response
Answers to the Reviewers’ comments
We are very grateful to the Reviewers for their time and constructive comments on our manuscript. We have implemented their comments and suggestions and wish to submit a revised version of the manuscript for further consideration in Materials. Changes in the initial version of the manuscript are highlighted in yellow in the revised version. Below, we also provide a point-by-point response explaining how we have addressed each of the Reviewers’ comment.
Answers to the Reviewer #2:
Comment:
Check the catalyst efficiency. If so, is the recovery of the catalyst no problem? In Fig 7, it appears to be completely separated, is there no such problem?
Authors' Response:
Thank you for all your suggestions. After the oligomerization process, isolating the oligomer and washing it with a mixture of methanol and hydrochloric acid, we added 1 ml of water to the reaction mixture. The catalyst cannot be reused because after activation by MMAO-12 it reacts with this reagent, which changes its structure and after the oligomerization is completed it cannot be separated from MMAO-12. In Fig. 7 we proposed the mechanism of the reaction under study and in fact, in the final stage, it can be seen that the complex compound is separated from MMAO-12. The proposed mechanism is entirely theoretical and has not been confirmed by our research. In our research we were unable to separate the complex from MMAO-12. We have not yet developed a method that would make this possible.
Comment:
Is there any way to determine the exact molecular size? (Not measurement/it means Poltmerization process) If not, I think it is difficult to explain the difference from the existing catalyst system.
Authors' Response:
There is an option to find isomers among the oligomeric products. These isomers can have internal double bonds or branchings. The sample obtained after conducting an oligomerization consists of chains of oligomers with a different number of mers. Potential components can be searched for in the NIST library by comparing the mass spectrum from the experiment to them. However, this is not always the case. We also looked for potential solutions, but we did not find it. Another option is to computer simulate with semi-empirical methods of quantum chemistry, these are computational methods based on the Hartree-Fock formalism.

Reviewer 3 Report
Comments to authors' reply
The authors addressed most of my comments, and the English quality was improved. However, I believe that the conclusions are still inconsistent with the data presented in the manuscript, and the authors should be more careful when discussing the results and constructing their hypotheses. Some of these discussions include the vanadium complex structure's assertion, the vanadium ligand's oxidation state, the uncertainty of the catalytic activity calculation, and the mechanism shown in Figure 7. This manuscript should undergo a major revision addressing all these points before publication.
My specific comments to the authors' responses are highlighted in bold.
Authors' Response:
Thank you very much for these valuable comments. The aim of our work was to investigate the catalytic properties of oxovanadium(IV) complex compound, therefore we did not focus in detail on the application of the oligomer. We sincerely apologize for the linguistic errors. All typos have been corrected. Changes in the initial version of the manuscript are highlighted in yellow in the revised version.
The manuscript still contains some formatting/typos that need to be fixed. Please, find below some of the sentences that need to be fixed:
Page 3, line 104: double space in the sentence after 100oC.
Page 5, line 158: Theoretical data have been calcuted according to the following procedure
Page 11, line 256: could the authors define "spherical hindrance"?
Authors' Response:
According to the Reviewer's recommendation the appropriate changes have been made as follows:
"The entire solution was heated at 100 °C to reflux for 90 minutes until the mixture changed color. It took 30 minutes to cool down. Crystallization was carried out at room temperature. "
Could the authors provide the rationale for such a long crystallization time? It would be useful for the reader to have at least one sentence explaining the reason for a crystallization time of 1 month.
Authors' Response:
According to the Reviewer's recommendation a process flow diagram for the oligomerization system was added to the manuscript (Fig. 1.). The ordinary pressure was equal to 1013 hPa.
Please, use ambient pressure instead of ordinary pressure.
Authors' Response:
According to the Reviewer's recommendation IR spectrum has been added.
|
Wavenumber [cm-1] |
Type of vibration with function group |
|
3571 cm−1 |
v(OH) |
|
1665 cm−1 |
v(COO) of dipic |
|
1352 cm−1 |
v(COO) of dipic |
|
983 cm−1 |
V=O stretching frequency |
|
452 cm−1 |
stretching vibration of the V-N |
The authors need to add references for the assignments of the IR peaks shown in Table 2. Also, the sentence "After comparing the results in the table above, we can conclude that the synthesis of the compound was correct because the literature values and the values of elemental" should be written as "The characterization results presented in Tables 1 and 2 and their validation with theoretical calculations and literature data indicate that the vanadium complex synthesis was correct."
Comment:
Asserting the double bond location just from IR analysis seems too speculative, and additional analysis should be included to validate this hypothesis. The MS spectrum shown in Figure 1 suggests different products with the same m/z value, indicating isomers among the oligomeric products. These isomers can have internal double bonds or branches. The authors should include not only the m/z but also the match with some potential components relative to the NIST library. This data would help develop a reaction pathway consistent with the data presented in the manuscript.
Authors' Response:
We agree with the Reviewer's opinion. However, despite our efforts to find the spectra in the NIST library, we were unable to find them.
In this case, the authors should provide a reasonable explanation in the manuscript addressing this point. Also, the authors cannot assert the double bond position since IR data alone is insufficient.
Comment:
What is the uncertainty of the catalytic activity calculation? This measure would be important to provide a precise comparison with the previous values reported in the literature – catalytic activities shown in Table 5. I also suggest that the authors show data for the consumption of 2-propen-1-ol and provide the mass balance for the experiment. This would be important to validate the catalytic activity calculated in the manuscript. Is V4+ the only oxidation state present in the synthesized catalyst? XPS data would also help in this regard.
Authors' Response:
Thank you for your comment. The aim of the research was not to characterize the obtained oligomer, but to investigate the catalytic properties of the catalyst in the oligomerization process. We did not perform validation due to the need to perform repeated processes, which is associated with higher costs. According to the IR analysis, we are sure that V4+ is the only oxidation state in the product. XPS analysis is not possible due to the lack of equipment in educational institutions in which we work. However, we will take into account some valuable suggestions from the editor for future research.
IR data cannot be used to assert the oxidation state/binding environment of ligand metals -XPS data should be used in this regard, instead. If XPS is not possible, I suggest that the authors include a sentence on the manuscript explaining their assumptions since the manuscript does not explain the quantification of the number of mmole of V4+ used to calculate the catalytic activity (Ca). Was it calculated theoretically, or was it measured via ICP? Also, the authors do discuss the characterization of the synthesized oligomers in the manuscript, as shown in Figures 4, 5, and 6 and Tables 4 and 5. There are uncertainties in the calculations of the catalytic activity (Ca) that are not being accounted for in the absence of a mass balance calculation. In this case, the authors should address the lack of mass balance calculation and include a sentence stating that the calculation has uncertainties that can lead to misleading comparisons with the data presented in Table 6.
Comment:
The level of detail for the mechanism presented in Figure 4 does not match the level of detail of the experimental data discussed in the manuscript, and the conclusions for the initiation, propagation, and termination steps of the oligomerization processes are far too speculative. I recommend that the authors consider the MS data when developing a reaction pathway instead of a detailed reaction mechanism.
Authors' Response:
Thank you for your valuable comments. The mechanism we have created is a proposed mechanism based on literature, not research. We will put the appropriate annotation in the manuscript. We realize that in order to confirm the reaction mechanism, more spectral analyzes and additional research are needed.
The authors should clearly state in the manuscript that the proposed mechanism is based on the literature and not their data. In the abstract, the authors state that "The aspect that enriches this work is the mechanism of oligomerization of allyl alcohol which we have proposed.", which gives the impression that the mechanism shown in Figure 7 was developed based on the data presented on the manuscript. Also, when explaining the mechanism, the authors do not provide any references. References for the mechanism shown in Figure should be included in section 4.
Authors' Response:
The quality of the images have been improved. Figures have been reduced so that they do not exceed the margin.
The quality of the images seems to remain unchanged relative to the previous version of the manuscript.
Minor comments:
The m/z values have too many significant digits.
The catalytic activity (Ca) (Page 10, line 239) should use a period instead of a comma.

Author Response
Answers to the Reviewers’ comments
We are very grateful to the Reviewers for their time and constructive comments on our manuscript. We have implemented their comments and suggestions and wish to submit a revised version of the manuscript for further consideration in Materials. Changes in the initial version of the manuscript are highlighted in yellow in the revised version. Below, we also provide a point-by-point response explaining how we have addressed each of the Reviewers’ comment.
Answers to the Reviewer #3:
Comment:
The manuscript still contains some formatting/typos that need to be fixed. Please, find below some of the sentences that need to be fixed:
Page 3, line 104: double space in the sentence after 100oC.
Page 5, line 158: Theoretical data have been calcuted according to the following procedure
Page 11, line 256: could the authors define "spherical hindrance"?
Authors' Response:
Thank you very much for these valuable comments. We sincerely apologize for the linguistic errors. All typos have been corrected. Changes in the initial version of the manuscript are highlighted in yellow in the revised version. Instead of "spherical hindrance” there should be “steric barrier".
Comment:
Could the authors provide the rationale for such a long crystallization time? It would be useful for the reader to have at least one sentence explaining the reason for a crystallization time of 1 month.
Authors' Response:
According to the Reviewer‘s recommendation the appropriate information has been added as follows:
“Crystallization lasted so long as to prevent the introduction of impurities that could appear if the process was carried out under conditions of reduced temperature.”
Comment:
Please, use ambient pressure instead of ordinary pressure.
Authors' Response:
We sincerely apologize for the linguistic errors. According to the Reviewer‘s recommendation the appropriate changes have been made as follows:
The whole oligomerization process was done at ambient pressure (1013 hPa), at room temperature and in nitrogen air.
Comment:
The authors need to add references for the assignments of the IR peaks shown in Table 2. Also, the sentence "After comparing the results in the table above, we can conclude that the synthesis of the compound was correct because the literature values and the values of elemental" should be written as "The characterization results presented in Tables 1 and 2 and their validation with theoretical calculations and literature data indicate that the vanadium complex synthesis was correct."
Authors' Response:
According to the Reviewer‘s recommendation the appropriate changes have been made as follows:
The characterization results presented in Tables 1 and 2 and their validation with theoretical calculations and literature data indicate that the vanadium complex synthesis was correct.
According to the Reviewer‘s recommendation references for the assignments of the IR peaks shown in Table 2 have been added:
[41] Drzeżdżon, J.; Pawlak, M.; Matyka, N.; Sikorski, A.; Gawdzik, B.; Jacewicz, D. Relationship between Antioxidant Activity and Ligand Basicity in the Dipicolinate Series of Oxovanadium(IV) and Dioxovanadium(V) Complexes. Int. J. Mol. Sci. 2021, 22, 9886.
[42] Li, M.; Ding, W.; Smee, J.J.; Baruah, B.; Willsky, G.R.; Crans, D.C. Anti-diabetic effects of vanadium (III, IV, V)–chlorodipicolinate complexes in streptozotocin-induced diabetic rats. Biometals 2009, 22, 895–905
Comment:
In this case, the authors should provide a reasonable explanation in the manuscript addressing this point. Also, the authors cannot assert the double bond position since IR data alone is insufficient.
Authors' Response:
The IR, 1H NMR and 13C NNMR spectra confirm that the obtained oligomer contains a double bond and a hydroxyl group. The indicated sentence has been corrected:
“The results of the IR studies showed that the end product of the oligomerization contained a double bond and a hydroxyl group.”
Additionally, we have added an appropriate note in the revised version of the manuscript:
NMR and IR test results confirm the structure of the obtained oligomers consisting of linked units of allyl alcohol.
Comment:
IR data cannot be used to assert the oxidation state/binding environment of ligand metals -XPS data should be used in this regard, instead. If XPS is not possible, I suggest that the authors include a sentence on the manuscript explaining their assumptions since the manuscript does not explain the quantification of the number of mmole of V4+ used to calculate the catalytic activity (Ca). Was it calculated theoretically, or was it measured via ICP? Also, the authors do discuss the characterization of the synthesized oligomers in the manuscript, as shown in Figures 4, 5, and 6 and Tables 4 and 5. There are uncertainties in the calculations of the catalytic activity (Ca) that are not being accounted for in the absence of a mass balance calculation. In this case, the authors should address the lack of mass balance calculation and include a sentence stating that the calculation has uncertainties that can lead to misleading comparisons with the data presented in Table 6.
Authors' Response:
According to the Reviewer‘s recommendation the appropriate changes have been made as follows:
“The number of mmole of V4+ used to calculate the catalytic activity (Ca) was calculated theoretically. In these calculations there is the lack of mass balance calculation, thus the calculation has uncertainties that can lead to misleading comparisons with the data presented in Table 6.”
Comment:
The authors should clearly state in the manuscript that the proposed mechanism is based on the literature and not their data. In the abstract, the authors state that "The aspect that enriches this work is the mechanism of oligomerization of allyl alcohol which we have proposed.", which gives the impression that the mechanism shown in Figure 7 was developed based on the data presented on the manuscript. Also, when explaining the mechanism, the authors do not provide any references. References for the mechanism shown in Figure should be included in section 4.
Authors' Response:
According to the Reviewer‘s recommendation the appropriate changes have been made as follows:
ABSTRACT: The aspect that enriches this work is the proposed mechanism of oligomerization of allyl alcohol based on the literature.
- The proposed mechanism of the oligomerization reaction of 2-propen-1-ol catalyzed by [VO(dipic)(H2O)2] · 2 H2O + MMAO-12
According to the Reviewer‘s recommendation, the reference was added in the revised version of the manuscript:
[44] Drzeżdżon, J.; Chmurzyński, L.; Jacewicz, D. Geometric isomerism effect on catalytic activities of bis(oxalato) diaquochromates(III) for 2-chloroallyl alcohol oligomerization, J. Chem. Sci, 2018, 130, 1-7.
Comment:
The m/z values have too many significant digits.
Authors' Response:
Too many significant digits have been deleted in the MALDI-TOF-MS spectrum.
Comment:
The catalytic activity (Ca) (Page 10, line 239) should use a period instead of a comma.
Authors' Response:
According to the Reviewer's recommendation, the catalytic activity formula has been edited.

Round 3
Reviewer 2 Report
Deer author
I checked the answer.
We look forward to expanding research into various materials in the future.
Thank you
Author Response
Answers to the Reviewers’ comments
We are very grateful to the Reviewers for their time and constructive comments on our manuscript. We have implemented their comments and suggestions and wish to submit a revised version of the manuscript for further consideration in Materials.
Reviewer 3 Report
The authors addressed most of my comments, and I only have some minor comments, as noted below. Overall, the manuscript improved significantly after the revisions. One comment that I have is regarding the mechanism proposed. As I mentioned in the previous revisions, such a detailed mechanism would require a proper mechanistic study, which in my view, goes beyond the data presented in the manuscript.
Please find my comments on the authors' replies in bold.
Authors' Response:
Thank you very much for these valuable comments. We sincerely apologize for the linguistic errors. All typos have been corrected. Changes in the initial version of the manuscript are highlighted in yellow in the revised version. Instead of "spherical hindrance" there should be "steric barrier".
The manuscript still contains some typos:
Page 2, Line 49: "It has become popular to use metallocenes, e. complex compounds containing…"
Page 2, Line 73: "They are used to produce synthetic rubbers, elastomers, and polyethylene [31]."
Page 4, Line 123: "…of 0.12 cm-1 for solid, liquid, and gaseous samples…"
Page 6, Line 179 and180: "It has been proven that with IR spectroscopy, hydrogen bonding, protonation, and chain propagation can be investigated."
Authors' Response:
The IR, 1H NMR and 13C NNMR spectra confirm that the obtained oligomer contains a double bond and a hydroxyl group. The indicated sentence has been corrected:
"The results of the IR studies showed that the end product of the oligomerization contained a double bond and a hydroxyl group."
Additionally, we have added an appropriate note in the revised version of the manuscript:
NMR and IR test results confirm the structure of the obtained oligomers consisting of linked units of allyl alcohol.
The authors should also include references for the assignment of the peaks shown in Table 3.
m/z values still contain too many significant digits (see page 7, line 198 and page 8, lines 199 and 200).
Authors' Response:
According to the Reviewer's recommendation the appropriate changes have been made as follows:
ABSTRACT: The aspect that enriches this work is the proposed mechanism of oligomerization of allyl alcohol based on the literature.
- The proposed mechanism of the oligomerization reaction of 2-propen-1-ol catalyzed by [VO(dipic)(H2O)2] · 2 H2O + MMAO-12
According to the Reviewer's recommendation, the reference was added in the revised version of the manuscript:
[44] Drzeżdżon, J.; Chmurzyński, L.; Jacewicz, D. Geometric isomerism effect on catalytic activities of bis(oxalato) diaquochromates(III) for 2-chloroallyl alcohol oligomerization, J. Chem. Sci, 2018, 130, 1-7.
The last sentence of the introduction (Page 3, lines 91 and 92) should also be changed to address the above comment.
In their previous publication (reference 44), the authors state that the mechanism derived for the Cr complex was based on the well-reported mechanism of Ziegler-Natta catalysts. I suggest that the authors add a similar sentence to section 4 of the manuscript submitted to Materials and the same references used in their previous publication:
- Cossee P 1964 Ziegler-Natta catalysis I: mechanism of polymerization of α-olefins with Ziegler-Natta catalysts J. Catal. 3 80
- Corradini P, Guerra G and Cavallo L 2004 Do new century catalysts unravel the mechanism of stereocontrol of old Ziegler-Natta catalysts? Acc. Chem. Res. 37 231
- Allegra G 1971 Discussion on the mechanism of polymerization of α-olefins with Ziegler-Natta catalysts Macromol. Chem. Phys. 145 235
Also, assuming that the mechanism of 2-propen-1-ol oligomerization with the vanadium complex is the same as that for 2-chloro-2-propen-1-ol with a chromium complex is highly speculative since the chemistry of the catalyst and the nature of the reactants are different.
Minor comments:
The authors need to define DMSO on Page 4, line 140.
It would be helpful for the reader to have the structure of the vanadium complex as an insert in Figure 2.
Author Response
Answers to the Reviewers’ comments
We are very grateful to the Reviewers for their time and constructive comments on our manuscript. We have implemented their comments and suggestions and wish to submit a revised version of the manuscript for further consideration in Materials. Changes in the initial version of the manuscript are highlighted in yellow in the revised version. Below, we also provide a point-by-point response explaining how we have addressed each of the Reviewers’ comment.
Answers to the Reviewer #3:
Comment:
Page 2, Line 49: "It has become popular to use metallocenes, e. complex compounds containing…"
Page 2, Line 73: "They are used to produce synthetic rubbers, elastomers, and polyethylene [31]."
Page 4, Line 123: "…of 0.12 cm-1 for solid, liquid, and gaseous samples…"
Page 6, Line 179 and180: "It has been proven that with IR spectroscopy, hydrogen bonding, protonation, and chain propagation can be investigated."
Authors' Response:
Thank you very much for these valuable comments. We sincerely apologize for the linguistic errors. All typos have been corrected. Changes in the initial version of the manuscript are highlighted in yellow in the revised version.
Comment:
The authors should also include references for the assignment of the peaks shown in Table 3.
Authors' Response:
According to the Reviewer's recommendation references have been added in the revised version of the manuscript.
[43] Iio, K., Kobayashi, K., Matsunaga, M. Radical polymerization of allyl alcohol and allyl acetate. Polym. Adv. Technol. 2007, 18, 953-958.
[44] Sawada, H., Tanba, K. I., Oue, M., Kawase, T., Mitani, M., Minoshima, Y., Moriya, Y. Synthesis and properties of novel fluoroalkylated allyl alcohol oligomers. Polymer, 1994, 35, 4028-4030.
Comment:
M/z values still contain too many significant digits (see page 7, line 198 and page 8, lines 199 and 200).
Authors' Response:
According to the Reviewer's recommendation too many significant digits in m/z values have been removed in the revised version of the manuscript.
Comment:
The last sentence of the introduction (Page 3, lines 91 and 92) should also be changed to address the above comment.
In their previous publication (reference 44), the authors state that the mechanism derived for the Cr complex was based on the well-reported mechanism of Ziegler-Natta catalysts. I suggest that the authors add a similar sentence to section 4 of the manuscript submitted to Materials and the same references used in their previous publication:
Cossee P 1964 Ziegler-Natta catalysis I: mechanism of polymerization of α-olefins with Ziegler-Natta catalysts J. Catal. 3 80
Corradini P, Guerra G and Cavallo L 2004 Do new century catalysts unravel the mechanism of stereocontrol of old Ziegler-Natta catalysts? Acc. Chem. Res. 37 231
Allegra G 1971 Discussion on the mechanism of polymerization of α-olefins with Ziegler-Natta catalysts Macromol. Chem. Phys. 145 235
Also, assuming that the mechanism of 2-propen-1-ol oligomerization with the vanadium complex is the same as that for 2-chloro-2-propen-1-ol with a chromium complex is highly speculative since the chemistry of the catalyst and the nature of the reactants are different.
Authors' Response:
According to the Reviewer's recommendation the last sentence of the introduction has been changed in the revised version of the manuscript.
Line 91-93: The aspect that enriches this work is the proposed mechanism of oligomerization of allyl alcohol based on the literature.
According to the Reviewer's recommendation the additional references have been added in the revised version of the manuscript.
[47] Cossee, P. Ziegler-Natta catalysis I: mechanism of polymerization of α-olefins with Ziegler-Natta catalysts J. Catal. 1964, 3, 80.
[48] Corradini, P.; Guerra, G.; Cavallo, L. Do new century catalysts unravel the mechanism of stereocontrol of old Ziegler-Natta catalysts? Acc. Chem. Res. 2004, 37, 231.
[49] Allegra, G. Discussion on the mechanism of polymerization of α-olefins with Ziegler-Natta catalysts Macromol. Chem. Phys. 1971, 145, 235.
Comment:
The authors need to define DMSO on Page 4, line 140.
Authors' Response:
According to the Reviewer‘s recommendation the appropriate changes have been made as follows:
Line 141 “…..was dissolved in 1 mL of toluene and 1 mL of anhydrous DMSO (anhydrous dimethyl sulfoxide).”
Comment:
It would be helpful for the reader to have the structure of the vanadium complex as an insert in Figure 2.
Authors' Response:
According to the Reviewer‘s recommendation Figure 2 with the structure of the precatalyst was added to the revised version of the manuscript.
Fig. 2. Chemical structure of [VO(dipic)(H2O)2] · 2 H2O.

This manuscript is a resubmission of an earlier submission. The following is a list of the peer review reports and author responses from that submission.
Round 1
Reviewer 1 Report
Introduction
- Page 2 Line no 45, “However, more and more often in publications, authors write about polymers or oligomers not only in meaning using it in the production of car tires [1, 2], packaging [3, 4], foil [5, 6] or oil [7, 8], but also polyolefins are used to production of medical implants [9, 10], anti‑HIV therapy (Human Immunodeficiency Virus) [11, 12], green chemistry [12‑14] and Alzheimer's treatment [15, 16]”
may be written as
“However, more and more often in publications, authors write about polymers or oligomers not only in meaning using them in the production of car tires [1, 2], packaging [3, 4], foil [5, 6] or oil [7, 8], but also polyolefins are used in the production of medical implants [9, 10], anti‑HIV therapy (Human Immunodeficiency Virus) [11, 12], green chemistry [12‑14] and Alzheimer's treatment [15, 16].”
- Page 2 Line no 49, “The synthesis of polymers requires special conditions for this, therefore catalysts are used, which lower the activation energy and speed up the process [17‑23]” may be written as “The synthesis of polymers requires special conditions therefore/for this, catalysts are used, which lower the activation energy and speed up the process [17‑23].”
- Page 2 Line no 57, “Modified methyl aluminoxane as activator oxidizes very quickly when there is oxygen in the reaction system, therefore nitrogen is introduced to prevent this” may be written as “Modified methyl aluminoxane, as an activatoroxidizes very quickly when there is oxygen in the reaction system, therefore nitrogen is introduced to prevent this.”
- Page 2 Line no 65, “Oxovanadium(IV) dipicolinate complex compound in its structure contains dipic (dipicolinate anion)which acts as a tridentate ligand” may be written as “Oxovanadium(IV) dipicolinate complex compound, in its structure, contains dipic (dipicolinate anion), which acts as a tridentate ligand.”
- Page 2 Line no 69, “it is used to remove corrosion, decontaminate nuclear reactors, and takes part in biological processes as a carrier of electrons and medical bioimaging [29, 30]” may be written as “it is used to remove corrosion, decontaminates nuclear reactors, and takes part in biological processes as a carrier of electrons and medical bioimaging [29, 30].”
- Page 2 Line no 72, “Oxovanadium(IV) compounds are used as precatalysts in the polymerization of olefins 72 due to their high catalytic activity and the quality of the products obtained” may be written as “ Oxovanadium(IV) compounds are used as precatalysts in the polymerization of olefins 72 due to their high catalytic activity and the quality of the products obtained.”
- Page 2 Line no 74, “They are used to produce synthetic rubbers, elastomers and polyethylene [31]” may be written as “They are used to produce synthetic rubbers, elastomers and polyethylene [31].”
- Page 2 Line no 75, “However, in our case, special attention was drawn to the dipicolinate complex of oxovanadium(IV) due to its widely described physicochemical and biological properties such as combating diabetes type I and II [32], cell metabolism [33], antioxidant properties [34], plasmid DNA cleavage, chromosomal aberrations and use in anticancer therapy [35]” may be written as “However, in our case, special attention was given to the dipicolinate complex of oxovanadium(IV), due to its widely described physicochemical and biological properties such as combating diabetes type I and II [32], cell metabolism [33], antioxidant properties [34], plasmid DNA cleavage, chromosomal aberrations and use in anticancer therapy [35].”
- Page 2 Line no 79, “In this publication, for the first timethe dipicolinate complex of oxovanadium(IV) is presented as a new precatalyst for an olefin oligomerization” may be written as “In this publication, for the first time, the dipicolinate complex of oxovanadium(IV) is presented as a new precatalyst for an olefin oligomerization.”
- Page 2 Line no 82, “The oligomerization reaction products were also analyzed using mass spectrometry techniques as matrix‑assisted laser desorption/ionization‑time‑of‑flight‑mass spectrometry (MALDI‑TOF‑MS), infrared spectroscopy (IR) and nuclear magnetic resonance (NMR)” may be written as “The oligomerization reaction products were also analyzed using mass spectrometry techniques such as matrix‑assisted laser desorption/ionization‑time‑of‑flight‑mass spectrometry (MALDI‑TOF‑MS), infrared spectroscopy (IR) and nuclear magnetic resonance (NMR).”
Material and Methods
- Page 3 Line no 96, “2 Dipicolinate oxovanadium(IV) complex synthesis: Aqueous vanadyl acetylacetonate (VO(acac)2) (2.13 mmol, 0.57 g) was added to 96 dipicolinic acid (H2dipic) (2.15 mmol, 0.36 g)” may be written as “2.2 Dipicolinate oxovanadium(IV) complex synthesis: Aqueous vanadyl acetylacetonate (VO(acac)2) (2.13 mmol, 0.57 g) was added to 96 dipicolinic acid (H2dipic) (2.15 mmol, 0.36 g).”
- Page 3 Line no 109, “4 IR spectra: The examination of the oligomerization product by infrared spectroscopy (IR) was performed in the range from 4000 cm‑1 to 600 cm‑1on a KBr pastil. The measurement was carried out on a Bruker IFS 66 spectrometer” may be written as “2.4 IR spectra: The examination of the oligomerization product by infrared spectroscopy (IR) was performed in the range from 4000 cm‑1 to 600 cm‑1 on a KBr pastil. The measurement was carried out on a Bruker IFS 66 spectrometer.”
- Page 3 Line no 115, “5 MALDI-TOF-MS spectra: Molecular weights of the 2‑propen‑1‑ol oligomer chains were determined using MALDI‑TOF‑MS spectrometer from the Bruker Biflex III company. 2,5‑Dihydroxybenzoic acid (DHB) has been used as matrix” may be written as “2.5 MALDI-TOF-MS spectra: Molecular weights of the 2‑propen‑1‑ol oligomer chains were determined using MALDI‑TOF‑MS spectrometer from the Bruker Biflex III company. 2,5‑Dihydroxybenzoic acid (DHB) was used as a matrix.”
- Page 3 Line no 120, “6 Nuclear magnetic resonance spectra (NMR): The measurement was carried out at 25 °C. The measurement frequency was 125.76 MHz for 13C NMR and 500.13 MHz for 1H NMR” may be written as “2.6 Nuclear magnetic resonance spectra (NMR): The measurement was carried out at 25 °C. The measured frequency was 125.76 MHz for 13C NMR and 500.13 MHz for 1H NMR.”
- Page 4 Line no 127, “7 The oligomerization process: The solution was then mixed with a magnetic stirrer. In the next step the following reagents were added: The entire oligomerization process was carried out under normal pressure, at room temperature and in nitrogen atmosphere” may be written as “2.7 The oligomerization process: The solution was then mixed with a help of magnetic stirrer. In the next step, the following reagents were added: The whole oligomerization process was done under ordinary pressure, at room temperature and in nitrogen air.”
- Page no 3 Line no 90, 95, 102, 103, 107, 112, 117,123, Headings from 2.1 to 2.7 as well as their explanation (lack equal tab and paragraph spacing) are not symmetrical and spacings between the number and the first word is not equal in all headings.
- Page 4 Line no 132-133, Check line spacing.
Results and discussion
- Page 4 Line no 143, “The synthesized complex was used to 2‑propen‑1‑ol oligomerization” may be written as “The synthesized complex was used for 2‑propen‑1‑ol oligomerization.”
- Page 4 Line no 148, “Using MALDI‑TOF‑MS method we characterized certain peaks thus allowing the identification of the number of units present in the oligomer chains using [VO(dipic)(H2O)2] • 2 H2O as a precatalyst” may be written as “ Using MALDI‑TOF‑MS method, we characterized certain peaks, thus allowing the identification of the number of units present in the oligomer chains using [VO(dipic)(H2O)2] • 2 H2O as a precatalyst.”
- Page 4 Line no 155, “The presence of oligomer chains of a specific length was confirmed by using mass 155 spectrometry” may be written as “ The presence of oligomer chains of a specific length was confirmed by using mass 155 spectrometry.”
- Page no 5 Line no 156, “peaks on the spectra have been assigned to the appropriate units spectra formed in the process of 2‑propen‑1‑ol oligomerization catalyzed by [VO(dipic)(H2O)2] • 2 H2O. The 649.999 m/z peak is derived from 2,5‑dihydroxybenzoic acid – matrix and molecular peak was identified with a mass/charge ratio of 703.937 m/z which contains twelve 2‑propen‑1‑ol units” may be written as “The appropriate units were assigned to the peaks in the spectra formed in the process of 2‑propen‑1‑ol oligomerization catalyzed by [VO(dipic)(H2O)2] • 2 H2O. The 649.999 m/z peak was (line no. 158, page 5) derived from 2,5‑dihydroxybenzoic acid – matrix and molecular peak was identified with a mass/charge ratio of 703.937 m/z which contained 2‑propen‑1‑ol 12 units.”
- Page 5 Line no 162, “Structure analysis of 2‑propen‑1‑ol oligomerization products has been conducted using nuclear magnetic resonance spectroscopic techniques. The 1H NMR spectrum is shown in Fig. 2 and the 13C NMR spectrum is shown in Fig. 3” may be written as “ analysis of 2‑propen‑1‑ol oligomerization products had been conducted using nuclear magnetic resonance spectroscopic techniques. The 1H NMR spectrum is shown in Fig. 2 and the 13C NMR spectrum is shown in Fig. 3.”
- Page 6 Line no 172, “In order to illustrate more precisely the structure of 2-propen-1-ol oligomers the Tables 3 and 4 based on the 1H NMR and 13C NMR spectra were prepared” may be written as “In order to illustrate the structure of 2-propen-1-ol oligomers more precisely, the Tables 3 and 4 based on the 1H NMR and 13C NMR spectra were prepared.”
- Page 6 Line no 174, “The peak values corresponding to specific carbon and hydrogen atoms depending on the type of spectrum have been highlighted” may be written as “The peak values correspond to specific carbon and hydrogen atoms depending on the type of spectrum have been highlighted.”
- Page 8 Line no 200, “Comparing the calculation results with the literature values, we conclude that the precatalyst [VO(dipic)(H2O)2] • 2 H2O belongs to the group of catalysts with high catalytic activity may be written as “Comparing the calculatedresults with the literature values, we concluded that the precatalyst [VO(dipic)(H2O)2] • 2 H2O belonged to the group of catalysts with high catalytic activity.”
- Page 8 Line no 205, “The mechanism of the oligomerization reaction of 2‑propen‑1‑ol catalyzed by [VO(dipic)(H2O)2] • 2 H2O with the participation of an activator (MMAO‑12) follows the mechanism of coordination polymerization (Fig. 4)” may be written as “The mechanism of the oligomerization reaction of 2‑propen‑1‑ol catalyzed by [VO(dipic)(H2O)2] • 2 H2O, with the participation of an activator (MMAO‑12), follows the mechanism of coordination polymerization (Fig. 4).”
- Page 8 Line no 209, “The approach of allyl alcohol to the center of oxovanadium(IV) with the participation of MMAO‑12 causes the formation of π complex between the alcohol's double bond and the active center.In the next step, coupling takes place between the terminal carbon atom and the active center of oxovanadium(IV), at the expense of the water molecule from the precatalyst, which migrates to the activator” may be written as “The approach of allyl alcohol to the center of oxovanadium(IV) with the participation of MMAO‑12 causes the formation of π complex between the alcohol's double bond and the active center. In the next step, coupling takes place between the terminal carbon atom and the active center of oxovanadium(IV), at the expense of the water molecule from the precatalyst, which migrates to the activator.”
- Page 8 Line no 251, “In the elimination stage, the obtained oligomer separates from the dipicolinate complex of oxovanadium(IV) and modified methylaluminoxane, through the migration of a water molecule from MMAO‑12 to the active center” may be written as “In the elimination stage, the obtained oligomer separates from the dipicolinate complex of oxovanadium(IV) and modifies methylaluminoxane, through the migration of a water molecule from MMAO‑12 to the active center.”
- Page 8 Line no 222, “The washing step is important because it could restore the stable structure of the catalyst under hydrolytic conditions” may be written as “The washing step is important because it can restore the stable structure of the catalyst under hydrolytic conditions.”
- Page 8 Line no 225, “Another fact confirming the correctness of the proposed termination step involved the experiment in which we added water to the reaction mixture after the oligomerization process as a result, we observed a free activator molecule precipitating out from a solution, suggesting that it is not bound to the catalyst anymore” may be written as “Another fact confirming the correctness of the proposed termination step, involved the experiment, in which we added water to the reaction mixture after the oligomerization process. As a result, we observed a free activator molecule precipitating out from a solution, suggesting that it was not bound to the catalyst anymore.”
- Page 4 Lines 135, 136, 137 have unequal line spacing and starting line 135 does not have tab spacing.
- Page 8 Line no 201 “Mention the calculated value of catalytic activity of pre-catalyst.”
- Line spacing of headings are not equal.
Tables
- Page 4 Line no 146, “Table 2: CH2” may be written as “Table 2: CH2”
- Page 4 Table no 2, “stretching vibration, stretching vibration, stretching vibration, bending vibrations” may be written as stretching vibrations, stretching vibrations, stretching vibrations, bending vibrations”
- Page 7 Table no 5, Try to adjust table 5 at one page.
- Font size of tables is not equal.
- Size of tables 3, 4 and 5 is not equal.
- Page 7 Table no 3-5, Align the peak values in the table 3 and 4 at the center.
- Page 7, Line no 188, Formula of catalytic activity is not symmetrical as well as formula values spacing is not equal.
- Table 5 catalytic activity formula is not clear.
- Page 4 Line no 138 and 146, Headings of table 1 and table 2 are not symmetrical to one another.
Figures
- Font size of figures is not equal.
- Size of figure 2 and 3 is not equal as well as their resolution is not good.
- Page 5,6 Line no 153,166,170, Headings of fig 1, fig 2 and fig 3 lack explanation of peaks’ information as well as axis information.
- Page 5 fig no 1, axes data is not mentioned.
- Page 10, Line no 234, In fig 4 try to give numbering to the steps of the reaction and do mention them in heading in one line i.e., Step 1 shows this, Step 2 shows that etc. Also write name of the compounds in the figure that are reacting. Also, do mention the propagation and termination steps on the figure.
Author Contributions
- Page 11 headings of lines 256, 257, 258, 260, 261, 262 are not symmetrical.
References
- Page 12,13 Line no 311, 312, 314, 318, 349 References no. 20, 21, 22 and 23 do not have tab spacing and reference no. 35 has extra tab spacing.
Additional Questions
- Page 4 table 2 shows IR spectrum of various functional groups and their different types of vibrations. What type of information does this spectrum gives? Also, what kind of information can be deduced from the functional groups present in the product? What is their role in oligomerization? It is not explained here in results.
- Page 5 fig 1, peaks formed at 15 and 18 units give which type of information? It is not explained in the results and discussion but mentioned in conclusion just in one line.
- Page 7 table 3 and 4 formed from fig 2 and 3 shows peak values of H and C atoms respectively. Which type of information is provided by the respective peak values? Explain them not give answer just in one word as given in table.
- What is the effect on oligomerization reaction if we replaced allyl alcohol with any other unsaturated alcohol? How can the efficiency of the process be affected?
Author Response
Answers to the Reviewers’ comments
We are very grateful to the Reviewers for their time and constructive comments on our manuscript. We have implemented their comments and suggestions and wish to submit a revised version of the manuscript for further consideration in the Journal. Changes in the initial version of the manuscript are highlighted in yellow in the revised version. Below, we also provide a point-by-point response explaining how we have addressed each of the Reviewers’ comment.
Answers to the Reviewer #1:
Comment:
Page 2 Line no 45, “However, more and more often in publications, authors write about polymers or oligomers not only in meaning using it in the production of car tires [1, 2], packaging [3, 4], foil [5, 6] or oil [7, 8], but also polyolefins are used to production of medical implants [9, 10], anti‑HIV therapy (Human Immunodeficiency Virus) [11, 12], green chemistry [12‑14] and Alzheimer's treatment [15, 16]”
may be written as “However, more and more often in publications, authors write about polymers or oligomers not only in meaning using them in the production of car tires [1, 2], packaging [3, 4], foil [5, 6] or oil [7, 8], but also polyolefins are used in the production of medical implants [9, 10], anti‑HIV therapy (Human Immunodeficiency Virus) [11, 12], green chemistry [12‑14] and Alzheimer's treatment [15, 16].”
Authors' Response:
According to the Reviewer‘s recommendation we changed the sentence constructions in the revised version of the manuscript as follows:
“However, more and more often in publications, authors write about polymers or oligomers not only in meaning using them in the production of car tires [1, 2], packaging [3, 4], foil [5, 6] or oil [7, 8], but also polyolefins are used in the production of medical implants [9, 10], anti‑HIV therapy (Human Immunodeficiency Virus) [11, 12], green chemistry [12‑14] and Alzheimer's treatment [15, 16].”
Comment:
Page 2 Line no 49, “The synthesis of polymers requires special conditions for this, therefore catalysts are used, which lower the activation energy and speed up the process [17‑23]” may be written as “The synthesis of polymers requires special conditions therefore/for this, catalysts are used, which lower the activation energy and speed up the process [17‑23].”
Authors' Response:
According to the Reviewer‘s recommendation the appropriate changes have been made:
The synthesis of polymers requires special conditions therefore, catalysts are used, which lower the activation energy and speed up the process [17‑23].
Comment:
Page 2 Line no 57, “Modified methyl aluminoxane as activator oxidizes very quickly when there is oxygen in the reaction system, therefore nitrogen is introduced to prevent this” may be written as “Modified methyl aluminoxane, as an activatoroxidizes very quickly when there is oxygen in the reaction system, therefore nitrogen is introduced to prevent this.”
Authors' Response:
According to the Reviewer‘s recommendation we changed the sentence constructions in the revised version of the manuscript:
“Modified methyl aluminoxane, as an activator oxidizes very quickly when there is oxygen in the reaction system, therefore nitrogen is introduced to prevent this.”
Comment:
Page 2 Line no 65, “Oxovanadium(IV) dipicolinate complex compound in its structure contains dipic (dipicolinate anion) which acts as a tridentate ligand” may be written as “Oxovanadium(IV) dipicolinate complex compound, in its structure, contains dipic (dipicolinate anion), which acts as a tridentate ligand.”
Authors' Response:
According to the Reviewer‘s recommendation the appropriate changes have been made:
“Oxovanadium(IV) dipicolinate complex compound, in its structure, contains dipic (dipicolinate anion), which acts as a tridentate ligand.”
Comment:
Page 2 Line no 69, “it is used to remove corrosion, decontaminate nuclear reactors, and takes part in biological processes as a carrier of electrons and medical bioimaging [29, 30]” may be written as “it is used to remove corrosion, decontaminates nuclear reactors, and takes part in biological processes as a carrier of electrons and medical bioimaging [29, 30].”
Authors' Response:
According to the Reviewer‘s recommendation we changed the sentence constructions in the revised version of the manuscript as follows:
“It is used to remove corrosion, decontaminates nuclear reactors, and takes part in biological processes as a carrier of electrons and medical bioimaging [29, 30].”
Comment:
Page 2 Line no 72, “Oxovanadium(IV) compounds are used as precatalysts in the polymerization of olefins 72 due to their high catalytic activity and the quality of the products obtained” may be written as “ Oxovanadium(IV) compounds are used as precatalysts in the polymerization of olefins 72 due to their high catalytic activity and the quality of the products obtained.”
Authors' Response:
According to the Reviewer‘s recommendation the appropriate changes have been made:
„Oxovanadium(IV) compounds are used as precatalysts in the polymerization of olefins due to their high catalytic activity and the quality of the products obtained.”
Comment:
Page 2 Line no 74, “They are used to produce synthetic rubbers, elastomers and polyethylene [31]” may be written as “They are used to produce synthetic rubbers, elastomers and polyethylene [31].”
Authors' Response:
According to the Reviewer‘s recommendation we changed the sentence constructions in the revised version of the manuscript as follows:
“They are used to produce synthetic rubbers, elastomers and polyethylene [31].”
Comment:
Page 2 Line no 75, “However, in our case, special attention was drawn to the dipicolinate complex of oxovanadium(IV) due to its widely described physicochemical and biological properties such as combating diabetes type I and II [32], cell metabolism [33], antioxidant properties [34], plasmid DNA cleavage, chromosomal aberrations and use in anticancer therapy [35]” may be written as “However, in our case, special attention was given to the dipicolinate complex of oxovanadium(IV), due to its widely described physicochemical and biological properties such as combating diabetes type I and II [32], cell metabolism [33], antioxidant properties [34], plasmid DNA cleavage, chromosomal aberrations and use in anticancer therapy [35].”
Authors' Response:
According to the Reviewer‘s recommendation the appropriate changes have been made:
“However, in our case, special attention was given to the dipicolinate complex of oxovanadium(IV), due to its widely described physicochemical and biological properties such as combating diabetes type I and II [32], cell metabolism [33], antioxidant properties [34], plasmid DNA cleavage, chromosomal aberrations and use in anticancer therapy [35].”
Comment:
Page 2 Line no 79, “In this publication, for the first time the dipicolinate complex of oxovanadium(IV) is presented as a new precatalyst for an olefin oligomerization” may be written as “In this publication, for the first time, the dipicolinate complex of oxovanadium(IV) is presented as a new precatalyst for an olefin oligomerization.”
Authors' Response:
According to the Reviewer‘s recommendation the indicated sentence has been corrected the revised version of the manuscript:
“In this publication, for the first time, the dipicolinate complex of oxovanadium(IV) is presented as a new precatalyst for an olefin oligomerization.”
Comment:
Page 2 Line no 82, “The oligomerization reaction products were also analyzed using mass spectrometry techniques as matrix‑assisted laser desorption/ionization‑time‑of‑flight‑mass spectrometry (MALDI‑TOF‑MS), infrared spectroscopy (IR) and nuclear magnetic resonance (NMR)” may be written as “The oligomerization reaction products were also analyzed using mass spectrometry techniques such as matrix‑assisted laser desorption/ionization‑time‑of‑flight‑mass spectrometry (MALDI‑TOF‑MS), infrared spectroscopy (IR) and nuclear magnetic resonance (NMR).”
Authors' Response:
According to the Reviewer‘s recommendation the appropriate changes have been made:
„The oligomerization reaction products were also analyzed using mass spectrometry techniques such as matrix‑assisted laser desorption/ionization‑time‑of‑flight‑mass spectrometry (MALDI‑TOF‑MS), infrared spectroscopy (IR) and nuclear magnetic resonance (NMR).”
Comment:
Page 3 Line no 96, “2 Dipicolinate oxovanadium(IV) complex synthesis: Aqueous vanadyl acetylacetonate (VO(acac)2) (2.13 mmol, 0.57 g) was added to 96 dipicolinic acid (H2dipic) (2.15 mmol, 0.36 g)” may be written as “2.2 Dipicolinate oxovanadium(IV) complex synthesis: Aqueous vanadyl acetylacetonate (VO(acac)2) (2.13 mmol, 0.57 g) was added to 96 dipicolinic acid (H2dipic) (2.15 mmol, 0.36 g).”
Authors' Response:
According to the Reviewer‘s recommendation we changed the sentence constructions in the revised version of the manuscript:
“2.2 Dipicolinate oxovanadium(IV) complex synthesis: Aqueous vanadyl acetylacetonate (VO(acac)2) (2.13 mmol, 0.57 g) was added to dipicolinic acid (H2dipic) (2.15 mmol, 0.36 g).”
Comment:
Page 3 Line no 109, “4 IR spectra: The examination of the oligomerization product by infrared spectroscopy (IR) was performed in the range from 4000 cm‑1 to 600 cm‑1on a KBr pastil. The measurement was carried out on a Bruker IFS 66 spectrometer” may be written as “2.4 IR spectra: The examination of the oligomerization product by infrared spectroscopy (IR) was performed in the range from 4000 cm‑1 to 600 cm‑1 on a KBr pastil. The measurement was carried out on a Bruker IFS 66 spectrometer.”
Authors' Response:
According to the Reviewer‘s recommendation the indicated part has been corrected as follows:
“2.4 IR spectra: The examination of the oligomerization product by infrared spectroscopy (IR) was performed in the range from 4000 cm‑1 to 600 cm‑1 on a KBr pastil. The measurement was carried out on a Bruker IFS 66 spectrometr.”
Comment:
Page 3 Line no 115, “5 MALDI-TOF-MS spectra: Molecular weights of the 2‑propen‑1‑ol oligomer chains were determined using MALDI‑TOF‑MS spectrometer from the Bruker Biflex III company. 2,5‑Dihydroxybenzoic acid (DHB) has been used as matrix” may be written as “2.5 MALDI-TOF-MS spectra: Molecular weights of the 2‑propen‑1‑ol oligomer chains were determined using MALDI‑TOF‑MS spectrometer from the Bruker Biflex III company. 2,5‑Dihydroxybenzoic acid (DHB) was used as a matrix.”
Authors' Response:
According to the Reviewer‘s recommendation we changed the sentence constructions in the revised version of the manuscript:
„2.5 MALDI-TOF-MS spectra: Molecular weights of the 2‑propen‑1‑ol oligomer chains were determined using MALDI‑TOF‑MS spectrometer from the Bruker Biflex III company. 2,5‑Dihydroxybenzoic acid (DHB) was used as a matrix.”
Comment:
Page 3 Line no 120, “6 Nuclear magnetic resonance spectra (NMR): The measurement was carried out at 25 °C. The measurement frequency was 125.76 MHz for 13C NMR and 500.13 MHz for 1H NMR” may be written as “2.6 Nuclear magnetic resonance spectra (NMR): The measurement was carried out at 25 °C. The measured frequency was 125.76 MHz for 13C NMR and 500.13 MHz for 1H NMR.”
Authors' Response:
According to the Reviewer‘s recommendation the appropriate changes have been made:
“2.6 Nuclear magnetic resonance spectra (NMR): The measurement was carried out at 25 °C. The measured frequency was 125.76 MHz for 13C NMR and 500.13 MHz for 1H NMR.”
Comment:
Page 4 Line no 127, “7 The oligomerization process: The solution was then mixed with a magnetic stirrer. In the next step the following reagents were added: The entire oligomerization process was carried out under normal pressure, at room temperature and in nitrogen atmosphere” may be written as “2.7 The oligomerization process: The solution was then mixed with a help of magnetic stirrer. In the next step, the following reagents were added: The whole oligomerization process was done under ordinary pressure, at room temperature and in nitrogen air.”
Authors' Response:
According to the Reviewer‘s recommendation we changed the sentence constructions in the revised version of the manuscript:
„2.7 The oligomerization process: The solution was then mixed with a help of magnetic stirrer. In the next step, the following reagents were added: The whole oligomerization process was done under ordinary pressure, at room temperature and in nitrogen air.”
Comment:
Page no 3 Line no 90, 95, 102, 103, 107, 112, 117,123, Headings from 2.1 to 2.7 as well as their explanation (lack equal tab and paragraph spacing) are not symmetrical and spacings between the number and the first word is not equal in all headings.
Authors' Response:
Editing errors have been corrected in the revised version of the manuscript.
Comment:
Page 4 Line no 132-133, Check line spacing.
Authors' Response:
Line spacing has been corrected.
Comment:
Page 4 Line no 143, “The synthesized complex was used to 2‑propen‑1‑ol oligomerization” may be written as “The synthesized complex was used for 2‑propen‑1‑ol oligomerization.”
Authors' Response:
According to the Reviewer‘s recommendation the appropriate changes have been made:
“The synthesized complex was used for 2‑propen‑1‑ol oligomerization.”
Comment:
Page 4 Line no 148, “Using MALDI‑TOF‑MS method we characterized certain peaks thus allowing the identification of the number of units present in the oligomer chains using [VO(dipic)(H2O)2] • 2 H2O as a precatalyst” may be written as “ Using MALDI‑TOF‑MS method, we characterized certain peaks, thus allowing the identification of the number of units present in the oligomer chains using [VO(dipic)(H2O)2] • 2 H2O as a precatalyst.”
Authors' Response:
According to the Reviewer‘s recommendation we changed the sentence constructions in the revised version of the manuscript:
“Using MALDI‑TOF‑MS method, we characterized certain peaks, thus allowing the identification of the number of units present in the oligomer chains using [VO(dipic)(H2O)2] ·2 H2O as a precatalyst.”
Comment:
Page 4 Line no 155, “The presence of oligomer chains of a specific length was confirmed by using mass 155 spectrometry” may be written as “ The presence of oligomer chains of a specific length was confirmed by using mass 155 spectrometry.”
Authors' Response:
According to the Reviewer‘s recommendation the appropriate changes have been made:
„The presence of oligomer chains of a specific length was confirmed by using mass spectrometry.„
Comment:
Page no 5 Line no 156, “peaks on the spectra have been assigned to the appropriate units spectra formed in the process of 2‑propen‑1‑ol oligomerization catalyzed by [VO(dipic)(H2O)2] • 2 H2O. The 649.999 m/z peak is derived from 2,5‑dihydroxybenzoic acid – matrix and molecular peak was identified with a mass/charge ratio of 703.937 m/z which contains twelve 2‑propen‑1‑ol units” may be written as “The appropriate units were assigned to the peaks in the spectra formed in the process of 2‑propen‑1‑ol oligomerization catalyzed by [VO(dipic)(H2O)2] • 2 H2O. The 649.999 m/z peak was (line no. 158, page 5) derived from 2,5‑dihydroxybenzoic acid – matrix and molecular peak was identified with a mass/charge ratio of 703.937 m/z which contained 2‑propen‑1‑ol 12 units.”
Authors' Response:
According to the Reviewer‘s recommendation we changed the sentence constructions in the revised version of the manuscript:
„The appropriate units were assigned to the peaks in the spectra formed in the process of 2‑propen‑1‑ol oligomerization catalyzed by [VO(dipic)(H2O)2] · 2 H2O. The 649.999 m/z peak was derived from 2,5‑dihydroxybenzoic acid – matrix and molecular peak was identified with a mass/charge ratio of 703.937 m/z which contained 2‑propen‑1‑ol 12 units.”
Comment:
Page 5 Line no 162, “Structure analysis of 2‑propen‑1‑ol oligomerization products has been conducted using nuclear magnetic resonance spectroscopic techniques. The 1H NMR spectrum is shown in Fig. 2 and the 13C NMR spectrum is shown in Fig. 3” may be written as “ analysis of 2‑propen‑1‑ol oligomerization products had been conducted using nuclear magnetic resonance spectroscopic techniques. The 1H NMR spectrum is shown in Fig. 2 and the 13C NMR spectrum is shown in Fig. 3.”
Authors' Response:
According to the Reviewer‘s recommendation the appropriate changes have been made:
“analysis of 2‑propen‑1‑ol oligomerization products had been conducted using nuclear magnetic resonance spectroscopic techniques. The 1H NMR spectrum is shown in Fig. 2 and the 13C NMR spectrum is shown in Fig. 3.”
Comment:
Page 6 Line no 172, “In order to illustrate more precisely the structure of 2-propen-1-ol oligomers the Tables 3 and 4 based on the 1H NMR and 13C NMR spectra were prepared” may be written as “In order to illustrate the structure of 2-propen-1-ol oligomers more precisely, the Tables 3 and 4 based on the 1H NMR and 13C NMR spectra were prepared.”
Authors' Response:
According to the Reviewer‘s recommendation the appropriate changes have been made:
„In order to illustrate the structure of 2-propen-1-ol oligomers more precisely, the Tables 3 and 4 based on the 1H NMR and 13C NMR spectra were prepared.”
Comment:
Page 6 Line no 174, “The peak values corresponding to specific carbon and hydrogen atoms depending on the type of spectrum have been highlighted” may be written as “The peak values correspond to specific carbon and hydrogen atoms depending on the type of spectrum have been highlighted.”
Authors' Response:
According to the Reviewer‘s recommendation the appropriate changes have been made:
„The peak values correspond to specific carbon and hydrogen atoms depending on the type of spectrum have been highlighted.”
Comment:
Page 8 Line no 200, “Comparing the calculation results with the literature values, we conclude that the precatalyst [VO(dipic)(H2O)2] • 2 H2O belongs to the group of catalysts with high catalytic activity may be written as “Comparing the calculatedresults with the literature values, we concluded that the precatalyst [VO(dipic)(H2O)2] • 2 H2O belonged to the group of catalysts with high catalytic activity.”
Authors' Response:
According to the Reviewer‘s recommendation the appropriate changes have been made:
“Comparing the calculated results with the literature values, we concluded that the precatalyst [VO(dipic)(H2O)2] · 2 H2O belonged to the group of catalysts with high catalytic activity.”
Comment:
Page 8 Line no 205, “The mechanism of the oligomerization reaction of 2‑propen‑1‑ol catalyzed by [VO(dipic)(H2O)2] • 2 H2O with the participation of an activator (MMAO‑12) follows the mechanism of coordination polymerization (Fig. 4)” may be written as “The mechanism of the oligomerization reaction of 2‑propen‑1‑ol catalyzed by [VO(dipic)(H2O)2] • 2 H2O, with the participation of an activator (MMAO‑12), follows the mechanism of coordination polymerization (Fig. 4).
Authors' Response:
According to the Reviewer‘s recommendation the appropriate changes have been made:
„The mechanism of the oligomerization reaction of 2‑propen‑1‑ol catalyzed by [VO(dipic)(H2O)2] · 2 H2O, with the participation of an activator (MMAO‑12), follows the mechanism of coordination polymerization (Fig. 4).”
Comment:
Page 8 Line no 209, “The approach of allyl alcohol to the center of oxovanadium(IV) with the participation of MMAO‑12 causes the formation of π complex between the alcohol's double bond and the active center.In the next step, coupling takes place between the terminal carbon atom and the active center of oxovanadium(IV), at the expense of the water molecule from the precatalyst, which migrates to the activator” may be written as “The approach of allyl alcohol to the center of oxovanadium(IV) with the participation of MMAO‑12 causes the formation of π complex between the alcohol's double bond and the active center. In the next step, coupling takes place between the terminal carbon atom and the active center of oxovanadium(IV), at the expense of the water molecule from the precatalyst, which migrates to the activator.”
Authors' Response:
According to the Reviewer‘s recommendation the appropriate changes have been made:
„The approach of allyl alcohol to the center of oxovanadium(IV) with the participation of MMAO‑12 causes the formation of π complex between the alcohol's double bond and the active center. In the next step, coupling takes place between the terminal carbon atom and the active center of oxovanadium(IV), at the expense of the water molecule from the precatalyst, which migrates to the activator.”
Comment:
Page 8 Line no 251, “In the elimination stage, the obtained oligomer separates from the dipicolinate complex of oxovanadium(IV) and modified methylaluminoxane, through the migration of a water molecule from MMAO‑12 to the active center” may be written as “In the elimination stage, the obtained oligomer separates from the dipicolinate complex of oxovanadium(IV) and modifies methylaluminoxane, through the migration of a water molecule from MMAO‑12 to the active center.”
Authors' Response:
According to the Reviewer‘s recommendation the appropriate changes have been made:
„In the elimination stage, the obtained oligomer separates from the dipicolinate complex of oxovanadium(IV) and modifies methylaluminoxane, through the migration of a water molecule from MMAO‑12 to the active center.”
Comment:
Page 8 Line no 222, “The washing step is important because it could restore the stable structure of the catalyst under hydrolytic conditions” may be written as “The washing step is important because it can restore the stable structure of the catalyst under hydrolytic conditions.”
Authors' Response:
According to the Reviewer‘s recommendation the appropriate changes have been made as follows:
„The washing step is important because it can restore the stable structure of the catalyst under hydrolytic conditions.”
Comment:
Page 8 Line no 225, “Another fact confirming the correctness of the proposed termination step involved the experiment in which we added water to the reaction mixture after the oligomerization process as a result, we observed a free activator molecule precipitating out from a solution, suggesting that it is not bound to the catalyst anymore” may be written as “Another fact confirming the correctness of the proposed termination step, involved the experiment, in which we added water to the reaction mixture after the oligomerization process. As a result, we observed a free activator molecule precipitating out from a solution, suggesting that it was not bound to the catalyst anymore.”
Authors' Response:
According to the Reviewer‘s recommendation the appropriate changes have been made:
„Another fact confirming the correctness of the proposed termination step, involved the experiment, in which we added water to the reaction mixture after the oligomerization process. As a result, we observed a free activator molecule precipitating out from a solution, suggesting that it was not bound to the catalyst anymore.”
Comment:
Page 4 Lines 135, 136, 137 have unequal line spacing and starting line 135 does not have tab spacing. Page 8 Line no 201 “Mention the calculated value of catalytic activity of pre-catalyst.” Line spacing of headings are not equal.
Authors' Response:
Editing errors and line spacing have been corrected.
Comment:
Page 4 Line no 146, “Table 2: CH2” may be written as “Table 2: CH2”
Authors' Response:
Error has been corrected.
Comment:
Page 4 Table no 2, “stretching vibration, stretching vibration, stretching vibration, bending vibrations” may be written as stretching vibrations, stretching vibrations, stretching vibrations, bending vibrations”
Authors' Response:
According to the Reviewer‘s recommendation the appropriate changes have been made:
„stretching vibrations, stretching vibrations, stretching vibrations, bending vibrations”
Comment:
Page 7 Table no 5, Try to adjust table 5 at one page.
Authors' Response:
Table no 5 has been adjust at one page.
Comment:
Font size of tables is not equal.
Authors' Response:
Front size of tabels has been corrected.
Comment:
Size of tables 3, 4 and 5 is not equal.
Authors' Response:
The size of tables 3, 4 and 5 has been corrected.
Comment:
Page 7 Table no 3-5, Align the peak values in the table 3 and 4 at the center.
Authors' Response:
Editing errors have been corrected.
Comment:
Page 7, Line no 188, Formula of catalytic activity is not symmetrical as well as formula values spacing is not equal.
Authors' Response:
Editing errors have been fixed.
Comment:
Table 5 catalytic activity formula is not clear.
Authors' Response:
Table 5 catalytic activity formula has been improved in the revised version of the manuscript:
Table 5. Catalyst efficiency classification based on their catalytic activity [39]
|
Catalyst efficiency |
Catalytic activity [g ∙ mmol-1 ∙ bar-1 ∙ h-1] |
|
Very low |
<1 |
|
Low |
1-10 |
|
Moderate |
10-100 |
|
High |
100-1000 |
|
Very high |
>1000 |
Comment:
Page 4 Line no 138 and 146, Headings of table 1 and table 2 are not symmetrical to one another.
Authors' Response:
Editing errors have been corrected.
Comment:
Font size of figures is not equal.
Size of figure 2 and 3 is not equal as well as their resolution is not good.
Page 5,6 Line no 153,166,170, Headings of fig 1, fig 2 and fig 3 lack explanation of peaks’ information as well as axis information.
Page 5 fig no 1, axes data is not mentioned.
Page 10, Line no 234, In fig 4 try to give numbering to the steps of the reaction and do mention them in heading in one line i.e., Step 1 shows this, Step 2 shows that etc. Also write name of the compounds in the figure that are reacting. Also, do mention the propagation and termination steps on the figure.
Authors' Response:
Editorial errors have been corrected. Explanations of the peaks are provided in the tables.
Reaction steps have been added. In our opinion, in the reaction mechanism, it is unnecessary to add the names of the compounds, because everything is exactly described in point 4.
Comment:
Page 11 headings of lines 256, 257, 258, 260, 261, 262 are not symmetrical.
Authors' Response:
Editorial errors have been corrected.
Comment:
Page 12,13 Line no 311, 312, 314, 318, 349 References no. 20, 21, 22 and 23 do not have tab spacing and reference no. 35 has extra tab spacing.
Authors' Response:
The format of references has been corrected.
Comment:
Page 4 table 2 shows IR spectrum of various functional groups and their different types of vibrations. What type of information does this spectrum gives? Also, what kind of information can be deduced from the functional groups present in the product? What is their role in oligomerization? It is not explained here in results.
Authors' Response:
According to the Reviewer‘s recommendation the relevant information has been added in the revised version of the manuscript:
“The absorption of infrared radiation is accompanied by changes in the vibrational energy of the molecules. Since this energy is quantized, only radiation with certain energies specific to the functional groups performing the vibrations is absorbed. It makes it possible to determine which functional groups are present in the analyzed sample. The condition for absorption of radiation is the change in the dipole moment of the molecule during the process. It has been proven that with IR spectroscopy, hydrogen bonding, protonation and chain propagation can be investigated, provided that aquization is used fast enough. The results of the IR studies show that the end product of the oligomerization contains a double bond located at one end of the chain.”
Comment:
Page 5 fig 1, peaks formed at 15 and 18 units give which type of information? It is not explained in the results and discussion but mentioned in conclusion just in one line.
Authors' Response:
According to the Reviewer‘s recommendation the relevant information has been added in the revised version of the manuscript:
“This is a confirmation that the obtained 2-propen-1-ol mixture the oligomers contained chains consisting of 12, 15 and 18 allyl alcohol units.”
Comment:
Page 7 table 3 and 4 formed from fig 2 and 3 shows peak values of H and C atoms respectively. Which type of information is provided by the respective peak values? Explain them not give answer just in one word as given in table.
Authors' Response:
According to the Reviewer‘s recommendation the relevant information has been added in the revised version of the manuscript:
“NMR spectroscopy is based on the observation of transitions between magnetic energy levels of the 1H hydrogen isotope in the case of 1H NMR. A lot of information about the structure of the molecule under study can be obtained from the NMR spectra. The number of signals provides information about the number of protons lying in the same environment. The intensity of the signals is proportional to the number of protons associated with this signal. On the other hand, the values of chemical shifts of signals in the spectrum depend on the environment in which the protons are located. The larger the peak, the stronger the coupling of the interaction between adjacent electron nuclei, the so-called spin-spin couplings. NMR test results confirm the structure of the obtained oligomers consisting of linked units of allyl alcohol”.
Comment:
What is the effect on oligomerization reaction if we replaced allyl alcohol with any other unsaturated alcohol? How can the efficiency of the process be affected?
Authors' Response:
According to the Reviewer‘s recommendation the relevant information has been added in the revised version of the manuscript:
“The highest catalytic activity in the research to date has been noticed with olefins containing -Cl in their structure for example 2-chloro-2-propen-1-ol. Thus achieving high purity and process efficiency could be increased. The use of ligands also plays an important role. Too extensive chains of organic clusters cause spherical hindrance, and thus low selectivity.”

Reviewer 2 Report
The paper is devoted to the olefin oligomerization using new precatalyst (oxovanadium(IV) dipicolinate complex [VO(dipic)(H2O)2] • 2 H2O) and the characterization of the oligomeric structure using IR, 1H and 13C NMR spectroscopy.
The manuscript can not be published in Materials depending on the following reasons:
1-Several issues seem to be unsolved. The structural characterization of the resulting compounds was not enlightened properly using 1H NMR and 13C NMR.
2- GPC is required to make a final decision for the polymeric structure.
3- The reactions should have been observed by changing some parameters, such as temperature, time, concentration…..
4- The presentation about figures needs to be improved.
Author Response
Answers to the Reviewer #2:
Comment:
1-Several issues seem to be unsolved. The structural characterization of the resulting compounds was not enlightened properly using 1H NMR and 13C NMR.
Authors' Response:
We apologize for these errors. NMR spectra were reanalyzed and their analysis was refined based on the literary data.
Comment:
2- GPC is required to make a final decision for the polymeric structure.
Authors' Response:
GPC studies has not been the aim of our research. The main aim of our research was to determine the catalytic activity of oxovanadium(IV) dipicolinate complex in the oligomerization of allyl alcohol.
Comment:
3- The reactions should have been observed by changing some parameters, such as temperature, time, concentration…..
Authors' Response:
Thank you very much for these valuable comments. We will certainly take this into account when planning further research, because the influence of parameters such as temperature, time, concentration may help to understand the characteristics of the oligomerization process.
Comment:
4- The presentation about figures needs to be improve.
Authors' Response:
According to the Reviewer‘s recommendation all figures have been improved.

Round 2
Reviewer 1 Report
No more comments, the authors revised the manuscript adequately.
Author Response
Thank you very much for the pertinent comments that contributed to the improvement of the quality of our manuscript
Reviewer 2 Report
There are still some doubts about the formation of oligomer.
Synthesis of oligomer is still doubtful.
1-The depicted IR and NMR spectra in the manuscript are not enough to confirm the structural characterization for the oligomer formation. Comparison is needed between the monomer's and oligomer's spectra.
2- The catalytic activity has to be controlled by changing some parameters, such as temperature, time, concentration .... etc.
Author Response
According to the Reviewer‘s recommendation the appropriate changes have been made:
Table 2. Characteristic IR spectrum absorption bands for the 2-propen-1-ol oligomerization product
|
Wavenumber of monomer [cm-1] |
Wavenumber of oligomer [cm-1] |
Type of vibration |
Function group |
|
3081 |
3425.48 |
stretching vibrations |
-OH |
|
2919 |
2991.75 |
stretching vibrations |
-CH |
|
1645 |
1650.52 |
stretching vibrations |
C=C |
|
1423 |
1438.01 |
bending vibrations |
-CH2 |
Table 3. Peak values and the corresponding hydrogen atoms derived from the 1H NMR spectrum for the products of 2-propen-1-ol oligomerization catalyzed by [VO(dipic)(H2O)2] • 2 H2O + MMAO-12
|
Peak value of monomer |
Peak value of oligomer |
Assigned hydrogen atoms |
|
5.991 |
6.04 |
CH2=CH- (oligomer) |
|
5.16 |
2.63 |
CH2=CH- |
|
4.148 |
1.71 |
HO-CH2- (oligomer) |
|
1.97 |
1.30 |
-OH (oligomer) |
Table 4. Peak values and the corresponding hydrogen atoms derived from the 13C NMR spectrum for the products of 2-propen-1-ol oligomerization catalyzed by [VO(dipic)(H2O)2] · 2H2O + MMAO-12
|
Peak value of monomer |
Peak value of oligomer |
Assigned carbons atoms |
|
118.2 |
70.79 |
HO-CH2-CH-CH2- (oligomer) |
|
133.5 |
70.76-70.31 |
HO-CH2-CH-CH2- (oligomer) |
|
65.6 |
37.64 |
-CH2-OH (oligomer) |
Comment:
2- The catalytic activity has to be controlled by changing some parameters, such as temperature, time, concentration .... etc.
Authors' Response:
We have clarified all parameters that are crucial for the oligomerization process so that the recording is correct.
The oligomerization process was carried out in a glass flask closed with a stopper. First, the precatalyst which was [VO(dipic)(H2O)2] · 2 H2O (3 μmol, 0.912 mg, 1.5 · 10-3 mol/dm3) was dissolved in 1 mL of toluene and 1 mL of DMSO. The solution was then mixed with a help of magnetic stirrer. In the next step the following reagents were added: 3 mL of MMAO-12 (modified methylaluminoxane, 7% aluminium in toluene, 7 wt. % in toluene) and 3 mL of 2-propen-1-ol. The whole oligomerization process was done under ordinary pressure (1.01325 bar), at room temperature (20 ℃) and in nitrogen air. After 90 minutes of oligomerization, a white gel was obtained and then washed with a mixture of 1 M hydrochloric acid and 1 M methanol in a 1:1 molar ratio.
We agree that along with a change in some parameters, e.g. temperature, it influences the catalytic activity of the complex. However, in accordance with the rules described in literature [39], we calculated the catalytic activity (Ca) for the measurement conditions in which we conducted the tests in accordance with the formula:
where:
m — mass of obtained oligomer [g]
n — number of mmoles of V4+ [mmol]
p — pressure [bar]
t — oligomerization time [h]
[39] Britovsek, G.J.; Gibson, V.C.; Wass, D.F. The search for new‐generation olefin polymerization catalysts: life beyond metallocenes, Angew.Chem. Int, 1999, 38, 428-447.
Due to the limited time and, above all, economic possibilities, we will be able to carry out additional research suggested in the review in the future, provided that financing is obtained.
